# Adaptive Time Encoding for Irregular Multivariate Time-Series Classification

**Sangho Lee**[1]**, Kyeongseo Min**[2]**, Youngdoo Son**[2][*]**, Hyungrok Do**[3]
[1]School of Industrial and Systems Engineering, Gyeongsang National University
[2]Department of Industrial and Systems Engineering, Dongguk University-Seoul
[3]Department of Population Health, NYU Grossman School of Medicine

## Abstract

Time series are often irregularly sampled with uneven time intervals. In multivariate cases, such irregularities may lead to misaligned observations across variables and varying observation counts, making it difficult to extract intrinsic patterns and degrading the classification performance of deep learning models. In this study, we propose an adaptive time encoding approach to address the challenge of irregular sampling in multivariate time-series classification. Our approach generates latent representations at learnable reference points that capture missingness patterns in irregular sequences, enhancing classification performance. We also introduce consistency regularization techniques to incorporate intricate temporal and intervariable information into the learned representations. Extensive experiments demonstrate that our method achieves state-of-the-art performance with high computational efficiency in irregular multivariate time-series classification tasks.

## 1   Introduction

Multivariate time series, which consist of multiple variables over time, are prevalent in diverse domains such as healthcare and finance [10, 14]. In practice, time series are often irregularly sampled with uneven time intervals between consecutive observations due to cost-saving measures, sensor

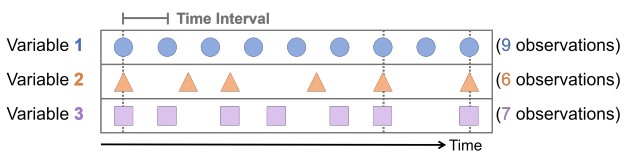

Figure 1: Example of irregular multivariate time series

failures, or medical interventions [4]. As shown in Figure 1, in multivariate cases, observations across different variables may not be aligned, and the number of observations in each variable can differ because different subsets of variables are recorded at each time point. These irregularities hinder the capture of intrinsic patterns, causing standard deep learning models, which assume that input sequences are regularly sampled, to perform poorly in classification tasks [16].

To address irregularities, previous work has developed deep learning models that directly learn from irregular time series to improve classification performance [2, 30]. In particular, recent studies have exploited attention mechanisms to focus on partial observations in irregular time series, successfully capturing complicated temporal patterns and achieving superior classification performance [3].

Despite their superiority, these methods encounter two notable limitations. First, they struggle to fully exploit missingness patterns inherent in uneven time intervals, which reflect the underlying processes that generate irregular time series [32]. For example, patients' examination schedules, which vary with changing health conditions or treatment needs, tend to reveal their health status. Second, they

---

[*]Corresponding author (`youngdoo@dongguk.edu`)

39th Conference on Neural Information Processing Systems (NeurIPS 2025).

often overlook intervariable relationships, which are particularly beneficial for classifying multivariate time series [8, 45]. Although some studies have considered these relationships as well as the challenge of irregular sampling [42, 47], they have usually used graph neural networks that require a high computational burden to capture intervariable dependencies [5, 41].

To address these issues, we design a novel interpolation-based encoder-classifier framework, *Adaptive Time Encoding Network* (ATENet), which learns reference time points and generates effective representations for irregular multivariate time-series classification. Specifically, we propose an *Adaptive Time Encoding* (ATE) that learns reference time points, which serve as queries in an attention mechanism in each training iteration, and creates fixed-length representations at the learned reference points. This approach reduces information loss by interpolating with unevenly spaced time points, effectively capturing missingness patterns from partially observed irregular time series. It also avoids the need for manual tuning of reference points. Moreover, we introduce temporal and intervariable consistency regularization terms to enhance representation quality by efficiently capturing intricate temporal patterns between consecutive observations and structural relations between variables, thereby boosting classification performance. Consequently, ATENet successfully classifies irregular multivariate time series by transforming them into fixed-length representations at the reference points that are adaptively learned to reflect both temporal and intervariable dependencies.

To demonstrate the effectiveness and efficiency of our method, we performed a series of experiments on irregular multivariate time-series classification. ATENet achieved superior classification performance with high computational efficiency compared to state-of-the-art (SOTA) methods.

This study has the following main contributions:

- We design a novel interpolation-based encoder-classifier framework that learns effective representations for irregular multivariate time-series classification;

- Our encoding approach directly learns reference points, rather than manually finding the optimal ones, to capture underlying patterns within irregular time series;

- We introduce temporal and intervariable consistency regularization terms to explicitly consider intricate temporal dynamics and relationships across variables;

- The proposed method achieved SOTA performance with high computational efficiency in irregular multivariate time-series classification.

## 2   Related Work

Irregular time series are characterized by uneven time intervals between adjacent observations. In multivariate cases, these irregularities mean that observations may not be aligned across variables, and the number of observations in each variable can also differ [12]. Such irregularities complicate the analysis for time series, often leading to poor classification performance of deep learning models.

A traditional approach to dealing with these irregularities is temporal discretization, which discretizes observations into consecutive and non-overlapping uniform bins [18, 22]. This approach is simple but requires additional handling for bins with more than one observation and leads to missing data when bins are empty [30].

As an alternative to temporal discretization, some previous studies intuitively preprocessed missing observations using various imputation or interpolation schemes and then fed them as regular sequences to standard deep learning models [23, 28]. However, the absence of observations can be informative on its own, making the regular imputation not always beneficial [1, 19]. Moreover, they can distort the inherent distribution of time series, leading to unintended distribution shifts [47].

Thus, several studies have developed deep learning models that directly learn from irregular time series to improve classification performance by preserving their intrinsic characteristics. For example, Che et al. [2] took the observed values and missing indicators as inputs for gated recurrent units and handled irregular time intervals through a decay mechanism. In addition, Wu et al. [40] focused on dynamically capturing temporal dependencies of irregular time series.

Some recent works have exploited attention mechanisms to successfully capture missingness patterns in irregular time series by considering all observations within the time series and finding informative ones [36, 48]. Horn et al. [9] incorporated an attention mechanism with differentiable set function

learning to handle misaligned observations across different variables. Shukla and Marlin [30] introduced a multi-time attention mechanism to learn temporal similarity from partial observations and generate continuous-time embeddings. However, they struggle to fully exploit missingness patterns that underlie irregular time series. Moreover, they do not explicitly capture intervariable dependencies, which provide rich information in multivariate time-series classification [8, 45].

While some studies have attempted to reflect intervariable relationships along with irregularities [42, 46, 47, 49], their computational complexities are extremely high due to the use of graph neural networks or additional complex attention mechanisms to capture intervariable dependencies [5, 41].

# 3 Proposed Method

We propose a novel interpolation-based encoder-classifier framework, *ATENet*, for irregular multivariate time-series classification. The encoder directly takes irregular sequences as inputs and generates their representations at learnable reference points, and the classifier predicts their class labels. In particular, a novel time encoding approach, *ATE*, is introduced to learn effective reference time points for capturing missingness patterns of irregular multivariate time series and generate latent representations at these reference points. Additionally, we introduce temporal and intervariable consistency regularization techniques to incorporate intricate temporal dynamics and relationships across variables into the representations, thereby enhancing classification performance. Figure 2 illustrates an overview of ATENet.

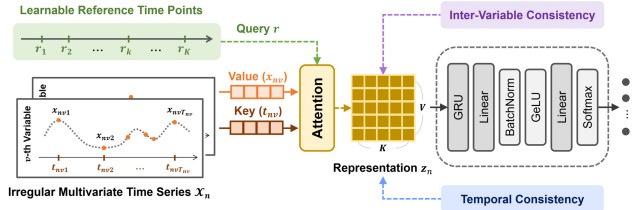

Figure 2: Overview of the proposed method

## 3.1 Problem Statement

Let $\mathbb{D} = \{(\mathcal{X}_n, \boldsymbol{y}_n)\}_{n=1}^{N}$ be a set of $N$ labeled samples, where $\mathcal{X}_n$ is an irregular multivariate time series, and $\boldsymbol{y}_n \in \{0, 1\}^C$ is its one-hot encoded label vector. The observations in each variable $v \in \{1, \cdots, V\}$ of $\mathcal{X}_n$ are irregularly recorded at different time points; hence, the number of observations $T_{nv}$ can differ across variables. Thus, for each variable $v$, we denote $\mathcal{X}_{nv} \in \mathcal{X}_n$ as a tuple $(\boldsymbol{t}_{nv}, \boldsymbol{x}_{nv})$, where $\boldsymbol{t}_{nv} = \{t_{nv1}, \cdots, t_{nvT_{nv}}\}$ and $\boldsymbol{x}_{nv} = \{x_{nv1}, \cdots, x_{nvT_{nv}}\}$ are the sets of the observed time points and values, respectively.

Given a set of learnable reference time points $\boldsymbol{r} = \{r_1, \cdots, r_K\}$, we define an encoder $f : \mathcal{X}_n \to \boldsymbol{z}_n$ and a classifier $g : \boldsymbol{z}_n \to \hat{\boldsymbol{y}}_n$, where $\boldsymbol{z}_n \in \mathbb{R}^{V \times K}$ is $V$-dimensional representations for each of $K$ reference points, and $\hat{\boldsymbol{y}}_n \in \mathbb{R}^C$ denotes the softmax probabilities for each class $c \in \{1, \cdots, C\}$ with respect to $\mathcal{X}_n$. Note that the dimension of $\boldsymbol{z}_{nk} \in \boldsymbol{z}_n$ is set equal to the number of variables $V$ of $\mathcal{X}_n$ to enforce intervariable consistency between $\mathcal{X}_n$ and $\boldsymbol{z}_n$. Our objective is to optimize $f$, $g$, and $\boldsymbol{r}$ to learn effective representations for irregular multivariate time-series classification. In the remainder of this paper, we omit the sample index $n$ for brevity when the context is clear.

## 3.2 Adaptive Time Encoding

In the encoder $f$, we introduce ATE that transforms an irregular multivariate time series $\mathcal{X}$ into a representation $\boldsymbol{z}$ on learnable reference points. To enhance representation quality, two consistency regularization techniques for reflecting temporal and intervariable dependencies are also suggested.

### 3.2.1 Learnable Reference Time Points

Previous studies with attention mechanisms generally transform an irregular sequence into a fixed-length representation by interpolating with regular time intervals [15, 30]. However, their representation may be insufficient to substitute the input sequence due to information loss caused by disregarding its irregularity in time intervals, and their classification performance tends to highly depend on the choice of reference points [31, 50]. Thus, we explicitly learn the reference points to

allow uneven time intervals and adaptively represent the partial observations of the irregular sequence without manually exploring the optimal reference points.

Specifically, we first learn a globally shared set of reference time points, which are not fixed but jointly optimized with model parameters to capture task-relevant temporal structures across the training data. These reference points serve as soft anchors that reflect representative temporal patterns, such as common event timings, even when sequences are irregular or misaligned. When the training data exhibit diverse alignment distributions, the learned reference points are flexibly positioned to reflect such variations. Importantly, although the reference points are shared, the attention-based interpolation is computed individually for each sample, conditioned on its actual observations. This design enables the model to adaptively align each sequence with the learned temporal structure, effectively handling variability in observation frequencies or event timings. Moreover, this design choice is motivated by three key benefits compared to using fully individualized reference points:

- *Robustness and stability*: It can mitigate sensitivity to per-sample noise and outliers.
- *Generalization*: It captures dataset-level temporal structure while retaining per-sample adaptability through attention-based interpolation.
- *Efficiency*: It reduces computational overhead compared to per-sample optimization without losing flexibility.

A detailed comparison between globally shared and fully individualized reference points is provided in Appendix G.

Let $r = \{r_1, \cdots, r_K\}$ be a query parameter that is a globally shared set of learnable reference time points uniformly initialized from zero to one. Note that during model training, this query parameter $r$ is optimized in an end-to-end manner. Given a $v$-th variable $\mathcal{X}_v = (t_v, x_v) \in \mathcal{X}$, our approach takes a query time point $r_k \in r$ and a set of keys and values, $t_v$ and $x_v$, as an input of the encoder $f$ and then obtains $V$-dimensional representations $z_k$ at $r_k$.

Following Shukla and Marlin [30], we first derive a time embedding vector of size $L$ for $t_{v\tau} \in t_v$ using a set of $H$ time embedding functions $\Phi = \{\phi_1, \cdots, \phi_H\}$. Two popular time embedding functions are:

1. Sinusoidal embedding function [38]:

$$\phi_h(t_{v\tau})[\ell] = \begin{cases} \sin(t_{v\tau}/T^{2\ell/L}), & \text{if } \ell \text{ is even} \\ \cos(t_{v\tau}/T^{2\ell/L}), & \text{if } \ell \text{ is odd} \end{cases} \tag{1}$$

where $\ell \in \{1, \cdots, L\}$ is $\ell$-th embedding component for $t_{v\tau}$, and $T$ is the number of all possible observations when fully observed (maximum sequence length). This function is independent of $h$, deriving the same embedding vector for all $H$ time embedding functions.

2. Learnable embedding function [11]:

$$\phi_h(t_{v\tau})[\ell] = \begin{cases} w_{h\ell} \cdot t_{v\tau} + b_{h\ell}, & \text{if } \ell = 1 \\ \sin(w_{h\ell} \cdot t_{v\tau} + b_{h\ell}), & \text{otherwise} \end{cases} \tag{2}$$

where $w_{h\ell}$ and $b_{h\ell}$ are learnable parameters that represent the frequency and phase shift of the sine function, respectively. It captures non-periodic patterns over time when $\ell = 1$; otherwise, it captures periodic patterns.

Our approach is agnostic to this function; hence, its choice is treated as a hyperparameter.

Subsequently, we define interpolation weights based on an attention mechanism. Specifically, the interpolation weights $\kappa_h(r_k, t_{v\tau})$ are computed as a scaled inner product attention between the time embedding vectors $\phi_h(t_{v\tau})$ and $\phi_h(r_k)$, which corresponds to the actually observed time point $t_{v\tau} \in t_v$ and the reference point $r_k$, respectively, as follows:

$$\kappa_h(r_k, t_{v\tau}) = \frac{e^{\phi_h(r_k)\phi_h(t_{v\tau})^\top/\sqrt{\epsilon}}}{\sum_{\tau'}^{T_v} e^{\phi_h(r_k)\phi_h(t_{v\tau'})^\top/\sqrt{\epsilon}}}, \tag{3}$$

where $T_v$ denotes the number of observations in $\mathcal{X}_v$, and $\epsilon$ is a scaling parameter. Note that the same $\phi_h$ is equally applied to all variables. Then, we obtain an univariate time function for $\mathcal{X}_v$, $\psi_{hv}(r_k, \mathcal{X}_v)$,

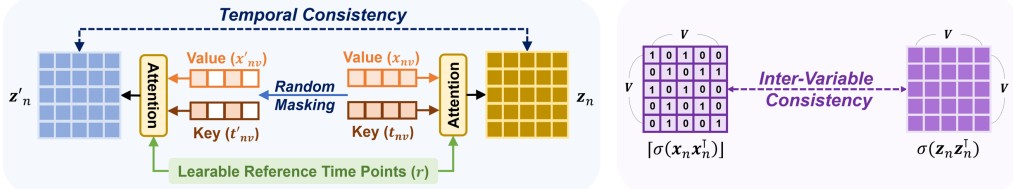

(a) Temporal consistency regularization  (b) Intervariable consistency regularization

Figure 3: Procedures for (a) temporal and (b) intervariable consistency regularization techniques

using the interpolation weights as follows:

$$\psi_{hv}(r_k, \mathcal{X}_v) = \sum_{\tau=1}^{T_v} \kappa_h(r_k, t_{v\tau}) \cdot x_{v\tau}, \tag{4}$$

where $t_{v\tau} \in \boldsymbol{t}_v$ and $x_{v\tau} \in \boldsymbol{x}_v$ denote an observed time point and value in $\mathcal{X}_v$. This function serves as a kernel smoothing for $\mathcal{X}_v$ [30]. Finally, for each reference point $r_k \in \boldsymbol{r}$, the representation $z_k \in \mathbb{R}^V$ is generated by a linear combination of the univariate time functions corresponding to $r_k$ across $V$ variables and $H$ heads. Formally, the $v'$-th embedding component of $z_k$, where $v' \in \{1, \cdots, V\}$, is calculated as follows:

$$z_k[v'] = \sum_{h=1}^{H} \sum_{v=1}^{V} \psi_{hv}(r_k, \mathcal{X}) \cdot W_{hvv'}, \tag{5}$$

where $W_{hvv'}$ are learnable parameters. Following these procedures for all reference points in parallel, we obtain the final representation $z = \{z_1, \cdots, z_K\}$ for $\mathcal{X}$.

### 3.2.2 Temporal Consistency Regularization

In general, time series contain information redundancy, easily enabling the recovery of missing observations through local temporal patterns obtained from adjacent observations. By hiding several observations of the time series with a high masking ratio, this information redundancy can be removed, thereby forcing the model to capture complex temporal relations [13]. Thus, we introduce a novel temporal consistency regularization term that exploits a masking technique to capture intricate temporal patterns in $\mathcal{X}$. Figure 3a illustrates the temporal consistency regularization technique.

Given $n$-th sample $\mathcal{X}_n = (\boldsymbol{t}_n, \boldsymbol{x}_n) \in \mathbb{D}$, we define a binary mask $\boldsymbol{m} \in \mathbb{R}^{V \times T}$, where $T$ is the maximum sequence length. Then, the masked keys and values, $\boldsymbol{t}'_n$ and $\boldsymbol{x}'_n$, are derived by element-wise multiplication of $\boldsymbol{m}$ with $\boldsymbol{t}_n$ and $\boldsymbol{x}_n$, respectively, as follows:

$$\boldsymbol{t}'_n = \boldsymbol{t}_n \circ \boldsymbol{m}, \ \boldsymbol{x}'_n = \boldsymbol{x}_n \circ \boldsymbol{m}. \tag{6}$$

We randomly pick the masking ratio $\in [0.1, 0.9]$ in every epoch to avoid the effort of finding the optimal masking ratio. Such random masking can encompass a variety of masking ratios, thereby enhancing the capability to capture sophisticated temporal patterns of $\mathcal{X}_n$. Consequently, we generate a masked context view $z'_n = \{z'_{n1}, \cdots, z'_{nK}\}$ of $\mathcal{X}_n$ by Eqs. (3)-(5) with $\boldsymbol{t}'_n$ and $\boldsymbol{x}'_n$.

To encourage temporal consistency, we employ both instance-wise and point-wise contrastive loss functions suggested by Yue et al. [44], which are complementary as they capture coarse-grained and fine-grained temporal dependencies, respectively. For $\mathcal{X}_n$, the instance-wise contrastive loss $\mathcal{L}_{TCI_n}$ is designed to maximize the similarity between the representation $z_{nk} \in z_n$ and its corresponding masked context view $z'_{nk} \in z'_n$, while minimizing the similarities to representations at the same reference point $r_k$ from other samples in the same batch. This loss is calculated as follows:

$$\mathcal{L}_{TCI_n} = -\frac{1}{K} \sum_{k=1}^{K} \log \frac{e^{z_{nk} \cdot z'_{nk}}}{\sum_{b=1}^{B} \left(e^{z_{nk} \cdot z'_{bk}} + \mathbb{1}_{[n \neq b]} e^{z_{nk} \cdot z_{bk}}\right)}, \tag{7}$$

where $B$ is the batch size, $K$ is the number of reference points, and $\mathbb{1}$ is the indicator function. In contrast, the point-wise contrastive loss $\mathcal{L}_{TCP_n}$, which uses the representations of $\mathcal{X}_n$ at different

reference points as negatives, is calculated by

$$\mathcal{L}_{TCP_n} = -\frac{1}{K} \sum_{k=1}^{K} \log \frac{e^{z_{nk} \cdot z'_{nk}}}{\sum_{k'=1}^{K} \left( e^{z_{nk} \cdot z'_{nk'}} + \mathbb{1}_{[k \neq k']} e^{z_{nk} \cdot z_{nk'}} \right)}, \tag{8}$$

Finally, the temporal consistency regularization term is defined as follows:

$$\mathcal{L}_{TC_n} = \frac{1}{2} \left( \mathcal{L}_{TCI_n} + \mathcal{L}_{TCP_n} \right). \tag{9}$$

### 3.2.3 Intervariable Consistency Regularization

Prior works that deal with intervariable relationships, which are known to be informative for multivariate time-series classification, have a high computational burden because they employed graph neural networks or incorporated further complicated attention mechanisms. Thus, to efficiently reflect the intervariable relationships, we exploit an intervariable consistency regularization term that is simply calculated based on the outer product. Figure 3b briefly displays the intervariable consistency regularization technique.

Given $\mathcal{X}_n = (t_n, x_n)$ and its representation $z_n$, we first define two outer product matrices for $x_n$ and $z_n$, denoted as $\mathcal{P}_n$ and $Q_n$, as follows:

$$\mathcal{P}_n = \lceil \sigma(x_n x_n^\top) \rfloor, \ Q_n = \sigma(z_n z_n^\top), \tag{10}$$

where $\sigma$ is the sigmoid function, and $\lceil \cdot \rfloor$ is the rounding operator. The dimensions of $\mathcal{P}_n$ and $Q_n$ are both $\|V\| \times \|V\|$. Following the outer product's properties, which capture the structural relations between two vectors [33], $\mathcal{P}_n$ and $Q_n$ can reflect intervariable dependencies in $x_n$ and $z_n$, respectively.

Then, we encourage intervariable consistency by employing the binary cross-entropy loss as follows:

$$\mathcal{L}_{VC_n} = \sum_{(p_{ij}, q_{ij}) \in (\mathcal{P}_n, Q_n)} p_{ij} \log q_{ij} + (1 - p_{ij}) \log(1 - q_{ij}), \tag{11}$$

where $p_{ij}$ and $q_{ij}$ are the $(i, j)$ elements of $\mathcal{P}_n$ and $Q_n$, respectively. Note that $\mathcal{P}_n$ is regarded as the ground truth of intervariable relations that should be maintained in the latent representation. Through this loss, we can efficiently capture intervariable dependencies, enriching the representation $z_n$.

### 3.3 Adaptive Time Encoding Network

ATENet is an end-to-end framework that sequentially combines an encoder configured by ATE with a classifier for irregular multivariate time-series classification. The encoder $f$ directly takes a labeled irregular sequence $(\mathcal{X}_n, y_n) \in \mathbb{D}$ as an input and generates a representation $z_n$ for the set of learnable reference time points $r$. The classifier $g$ then uses $z_n$ to predict the softmax probabilities $\hat{y}_n$.

### 3.3.1 Simple Classifier

If the encoder $f$ learns the representations that successfully substitute irregular multivariate time series through ATE, we can achieve high classification performance even with a simple classifier. Thus, we simply design the classifier $g$ as a gated recurrent unit followed by two fully connected layers, where the first layer includes batch normalization and a GeLU activation function.

Let $y_n = \{y_{n1}, \cdots, y_{nC}\}$ and $\hat{y}_n = \{\hat{y}_{n1}, \cdots, \hat{y}_{nC}\}$ be the one-hot encoded label vector and predicted softmax probabilities for an irregular multivariate time series $\mathcal{X}_n$, where $C$ is the number of classes. We define a classification loss as the cross-entropy combined with label smoothing, parameterized by $\eta$, to prevent overfitting of the model and improve its generalization performance [34], as follows:

$$\mathcal{L}_{CL_n} = -\sum_{c=1}^{C} \left( (1 - \eta) y_{nc} + \frac{\eta}{C} \right) \log \hat{y}_{nc}. \tag{12}$$

### 3.3.2 Optimization

Given $\mathbb{D} = \{(\mathcal{X}_n, y_n)\}_{n=1}^{N}$, we train $f$, $g$, and $r$ with the following loss function:

$$\mathcal{L} = \frac{1}{N} \sum_{n=1}^{N} \left( \mathcal{L}_{CL_n} + \alpha \mathcal{L}_{TC_n} + \beta \mathcal{L}_{VC_n} \right), \tag{13}$$

| Metric | Dataset | mTAND | DGM$^2$ | GRU-D | MTGNN | Transformer | Trans-mean | SeFT | Raindrop | Warpformer | MTSFormer | ATENet |
|---|---|---|---|---|---|---|---|---|---|---|---|---|
| AUROC | P12-M | 84.18±1.20 | 71.08±2.30 | 48.62±2.41 | 61.59±5.79 | 82.92±0.72 | 83.39±0.56 | 68.05±1.49 | 81.19±1.76 | 79.35±1.65 | 84.11±0.71 | **85.54**±1.26 |
|  | P12-L | 49.60±3.16 | 69.46±1.47 | 49.82±3.85 | 68.36±6.09 | 59.05±1.81 | 61.64±1.54 | 64.70±2.01 | 70.40±1.60 | 74.57±2.28 | 75.17±1.09 | **79.64**±2.24 |
|  | P19 | 80.00±1.23 | 81.96±2.05 | 87.16±1.34 | 85.07±3.54 | 77.56±3.06 | 78.57±3.02 | 77.89±2.62 | 85.93±2.24 | 85.41±2.39 | **88.96**±2.01 | 84.02±1.38 |
|  | PAM | 92.21±0.70 | 96.87±0.50 | 91.72±0.59 | 96.95±0.32 | 96.61±1.27 | 97.64±0.25 | 74.46±6.70 | 98.73±0.25 | 97.94±0.45 | 98.39±0.28 | **99.18**±0.15 |
| AUPRC | P12-M | 52.89±2.27 | 29.99±2.24 | 14.83±1.55 | 24.25±5.49 | 46.35±2.81 | 48.54±2.24 | 24.43±3.10 | 42.14±3.32 | 41.98±1.30 | 48.53±2.55 | **53.31**±2.02 |
|  | P12-L | 92.42±1.09 | 96.42±0.41 | 93.41±0.93 | 96.41±1.08 | 94.16±0.99 | 94.65±0.80 | 95.28±0.24 | 96.57±0.51 | 96.99±0.34 | 97.43±0.28 | **97.70**±0.38 |
|  | P19 | 31.24±4.15 | 31.12±5.25 | 47.37±2.97 | 41.13±8.01 | 29.60±6.26 | 28.05±6.23 | 30.34±1.80 | 50.63±3.32 | 41.12±3.30 | **57.96**±4.10 | 41.16±3.02 |
|  | PAM | 74.95±2.68 | 88.28±1.28 | 75.78±2.02 | 88.85±2.00 | 86.73±4.21 | 91.50±0.61 | 36.43±12.23 | 95.48±0.91 | 92.75±1.43 | 94.21±0.71 | **97.61**±0.26 |
| *Average Rank* |  | 7.50 | 6.88 | 8.25 | 6.75 | 8.13 | 6.63 | 9.13 | 3.75 | 4.63 | 2.38 | **2.00** |

Table 1: Classification performance of ATENet and baselines. The best score in each dataset is shown in bold.

where $\alpha$ and $\beta$ weight the temporal and intervariable consistency regularization terms, respectively.

In summary, $\mathcal{L}_{CL_n}$ allows the representations to be directly affected by class labels and capture discriminative features relevant to classification, while $\mathcal{L}_{TC_n}$ and $\mathcal{L}_{VC_n}$ capture inherent temporal patterns and intervariable relationships in irregular multivariate time series, thereby enhancing classification performance. Pseudo-code and complexity analysis are given in Appendices A and B.

## 4 Experiments

### 4.1 Experimental Settings

Here, we briefly describe the experimental settings. The implementation details and sensitivity analyses for hyperparameters are provided in Appendices D and I.

**Baselines.** We compared ATENet with 10 SOTA methods: *mTAND* [30], *DGM$^2$* [40], *GRU-D* [2], *MTGNN* [42], *Transformer* [38], *Trans-mean*, *SeFT* [9], *Raindrop* [47], *Warpformer* [46], and *MTSFormer* [49]. All baselines, except *Trans-mean*, are mentioned in Section 2; *Trans-mean* is a method that combines *Transformer* with average interpolation, which imputes missing observations by the average observed value of each variable.

**Datasets.** To validate our method, ATENet, we used three irregular multivariate time-series datasets:

- *P12* [7], which includes 11,988 patients recorded by 36 sensors in the first 48-hour stay in the intensive care unit, has two predictive binary class labels: in-hospital mortality (*P12-M*) and hospitalization length (*P12-L*).
- *P19* [27] contains 38,803 patients monitored by 34 sensors. Each sample is annotated with a binary class label for the occurrence of sepsis.
- *PAM* [25] has 5,333 samples for eight activities of daily life measured by 17 sensors.

Further details for each dataset are provided in Appendix C.

**Evaluation metrics.** We employed the area under the receiver operating characteristic curve (AUROC) and area under the precision-recall curve (AUPRC) to evaluate classification performance while considering the imbalance in each dataset. We repeated each experiment five times and reported the averages and standard deviations.

### 4.2 Experimental Results

#### 4.2.1 Classification Performance

Table 1 shows the averages and standard deviations of AUROC and AUPRC scores of the baselines and ATENet for each dataset. The results of statistical tests, which confirm the significance of the differences in classification performance, are given in Appendix E.

ATENet remarkably outperformed the baselines by achieving the best average rank of 2.00 across all datasets and metrics, demonstrating the effectiveness of ATENet for irregular multivariate time-series classification. Especially for the P12-M and P12-L datasets, which have the most variables, and the PAM dataset, which has the longest sequences, our method showed the highest classification performance in both AUROC and AUPRC scores. Moreover, ATENet performed significantly better than the baselines in most cases (see Table A2). In Appendix F, we further discuss these results.

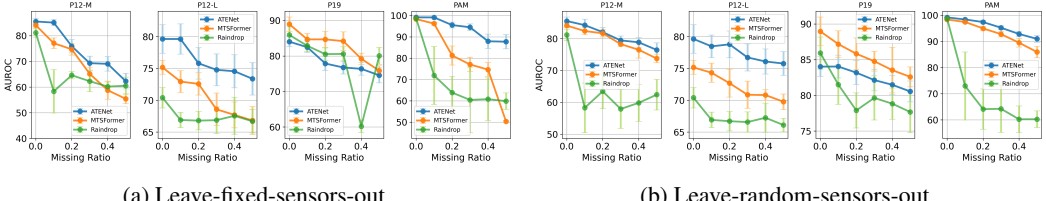

|  (a) Leave-fixed-sensors-out | (b) Leave-random-sensors-out |

Figure 4: AUROC scores of ATENet, MTSFormer, and Raindrop when dropping variables with various ratios ∈ [0.1, 0.5] in (a) *leave-fixed-sensors-out* and (b) *leave-random-sensors-out* scenarios

### 4.2.2 Robustness to Missing Variables

ATENet can mitigate a drastic performance drop even when some variables are missing by introducing intervariable consistency to capture the structural relations of inputs. We examined the robustness of ATENet by selecting a subset of variables and hiding their observations in the test set. Following Zhang et al. [47], we set up two scenarios:

- *Leave-fixed-sensors-out*: The most informative variables determined by information gain analysis are dropped [47]. The dropped variables are fixed across every sample.
- *Leave-random-sensors-out*: Missing variables are not fixed but are selected randomly from each sample.

In Figure 4, we compared the AUROC of ATENet and those of MTSFormer and Raindrop, which are the SOTA methods that showed the second-best and third-best performance in Table 1, under these scenarios where variables are removed at various ratios ranging from 0.1 to 0.5.

As shown in Figure 4a, ATENet showed more robust performance with low standard deviations than MTSFormer and Raindrop in the *leave-fixed-sensors-out* scenario. MTSFormer and Raindrop also take into account intervariable dependencies, but their performance remarkably declined on most datasets. Especially in the P19 dataset, Raindrop's performance notably dropped when 40% of variables were removed. In addition, in the PAM dataset, MTSFormer showed a performance drop of approximately 50% when 50% of variables were removed.

In the *leave-random-sensors-out* scenario, as exhibited in Figure 4b, the proposed method also showed significantly better robustness to missing variables than MTSFormer and Raindrop. For example, Raindrop's performance dropped by approximately 30% when more than 20% of variables were randomly dropped in each sample of the PAM dataset.

Therefore, this shows the robustness of ATENet to the absence of variables by successfully capturing intervariable relationships in irregular multivariate time series. The complete results, including a comparison of ATENet with all baselines, are provided in Appendix G.1.

Furthermore, our method can perform robustly when some observations are missing along the time axis by effectively capturing intricate temporal dependencies owing to learnable reference points and temporal consistency regularization. The results of these experiments are provided in Appendix H.

### 4.2.3 Computational Efficiency

The proposed method efficiently reflects intervariable relationships in irregular multivariate time series by solely computing the outer product of variables and that of representations. Figure 5 shows the number of parameters and processing time of ATENet with those of *MTGNN*, *Raindrop*, *Warpformer*, and *MTSFormer*, which achieved high classification performance among the baselines by considering structural relationships between variables (see Table 1).

We observed that ATENet is remarkably efficient compared to MTGNN and Raindrop, which require high computation complexities due to their use of graph neural networks, in terms of both the number of parameters and processing time. In particular, for all datasets, the proposed method requires at least 10 times fewer parameters and achieves speeds at least 3 times faster than Raindrop. Moreover, although the efficiency gains over Warpformer and MTSFormer are not as large as those over Raindrop and MTGNN, our method remains more efficient, as both rely on additional complicated attention mechanisms to capture intervariable dependencies.

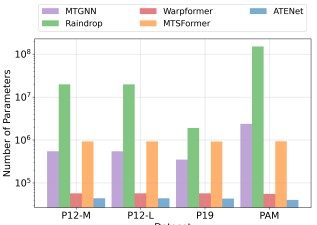

(a) Number of Parameters

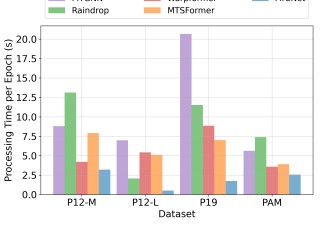

(b) Processing Time per Epoch (s)

Figure 5: (a) Number of parameters and (b) processing time per epoch for MTGNN, Raindrop, Warpformer, MTSFormer, and ATENet

Table 2: Classification performance of ATENet and ablation models

| Metric | Dataset | Regular | Sparse | Dense | ATENet |
|--------|---------|---------|--------|-------|--------|
| AUROC | P12-M | 85.49 | 85.61 | **85.65** | 85.54 |
|  | P12-L | 77.26 | 75.35 | 77.35 | **79.64** |
|  | P19 | 83.58 | 82.67 | 83.46 | **84.02** |
|  | PAM | 99.00 | 99.10 | 98.97 | **99.18** |
| AUPRC | P12-M | 51.26 | 51.22 | 51.20 | **53.31** |
|  | P12-L | 97.55 | 97.54 | 97.55 | **97.70** |
|  | P19 | 38.06 | 36.44 | 38.85 | **41.16** |
|  | PAM | 96.99 | 97.13 | 96.97 | **97.61** |

## 4.3 Ablation Studies

We investigated the effects of three key components of ATENet: *learnable reference time points*, *temporal consistency*, and *intervariable consistency*.

### 4.3.1 Learnable Reference Time Points

To examine the impact of learnable reference time points, we compared the classification performance of the proposed method against ATENet with fixed reference time points. We designed three ablation models where the reference time points are at regular time intervals (Regular), increasingly sparse time intervals (Sparse), and increasingly dense time intervals (Dense), respectively. In precise, their reference time points $r = \{r_1, \cdots, r_K\}$ were characterized as follows:

- Regular: $r_k = r_1 + (k-1), \forall k \in \{1, \cdots, K\}$

- Sparse: $r_k = r_1 e^{(k-1)}, \forall k \in \{1, \cdots, K\}$

- Dense: $r_k = r_1 e^{-(k-1)}, \forall k \in \{1, \cdots, K\}$

As presented in Table 2, our method outperformed three ablation models, showing the effectiveness of learnable reference points in classification performance.

Furthermore, in Figure 6, we visualized the distribution of observed time points across all training samples in the PAM dataset and illustrated the attention weights corresponding to the learnable and fixed (Regular) reference points using an example sequence from the same dataset. As shown in Figure 6a, the blue bars indicate the overall distribution of observed time points, while the learnable reference points (green circles) are predominantly located in regions with high observation density. In Figure 6b, the blue dots denote the observed values, and the learnable reference time points (green circles) adaptively align with clusters of these observations (brown circles), effectively capturing irregular sampling patterns and emphasizing informative observations through highly deviated attention weights. In contrast, Figure 6c shows that the Regular reference points fail to capture temporal irregularities and find informative time points for classification, as their attention weights are rather uniformly distributed. These results reaffirm the benefit of our approach in reflecting missingness patterns in irregular sequences and reducing information loss that may occur with uniformly spaced interpolation, leading to more effective representations for classification.

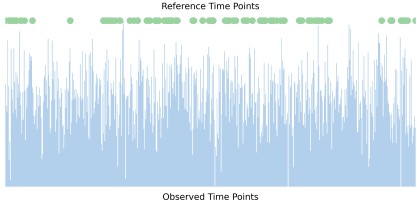

(a) Distribution of Observed Time Points

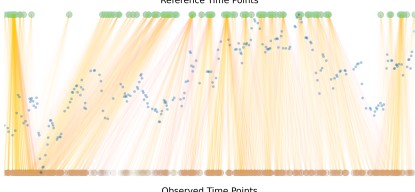

(b) Learnable Reference Time Points

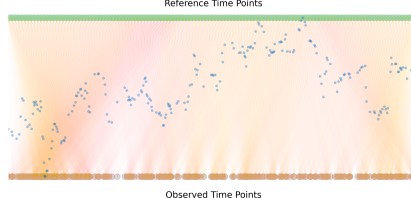

(c) Fixed Reference Time Points (Regular)

Figure 6: Visualization of (a) distribution of the observed time points and attention weights for (b) learnable reference points from ATENet and (c) fixed ones (Regular)

### 4.3.2 Temporal Consistency Regularization

To capture intricate temporal patterns, our method encourages temporal consistency by using instance-wise and point-wise contrastive loss functions, $\mathcal{L}_{TCI}$ and $\mathcal{L}_{TCP}$. As listed in Table 3, ATENet *w/o* $\mathcal{L}_{TC}$, ATENet *w/o* $\mathcal{L}_{TCP}$, and ATENet *w/o* $\mathcal{L}_{TCI}$ dropped the average performance compared to ATENet. Thus, we validated that temporal consistency regularization is useful for capturing temporal dependencies spanning various time intervals, thereby enhancing classification performance. The complete results are provided in Appendix G.2

| Metric | *w/o* $\mathcal{L}_{TC}$ | *w/o* $\mathcal{L}_{TCP}$ | *w/o* $\mathcal{L}_{TCI}$ | *w/o* $\mathcal{L}_{VC}$ |
|---|---|---|---|---|
| AUROC | 0.20 | 0.13 | 0.91 | 3.22 |
| AUPRC | 1.28 | 0.57 | 1.63 | 3.33 |

Table 3: Average performance drop rate (%) of ablation models without consistency regularization compared to ATENet

### 4.3.3 Intervariable Consistency Regularization

The proposed method efficiently captures intervariable relationships by ensuring intervariable consistency between inputs and their representations. As shown in Table 3, ATENet *w/o* $\mathcal{L}_{VC}$ highly dropped the average performance of 3.33% in the AUPRC compared to ATENet, demonstrating that this regularization term can provide rich information for accurate classification. The complete results for each dataset are provided in Appendix G.2. Furthermore, as

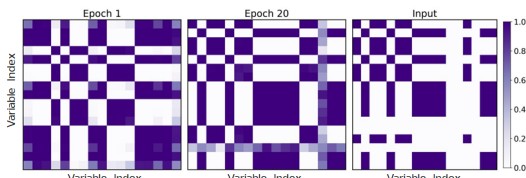

Figure 7: Visualization of intervariable relations from the input and from the learned representations at epochs 1 and 20

shown in Figure 7, the learned representation progressively aligns with the input structure during training, highlighting the efficacy of intervariable consistency regularization in reflecting intervariable information.

## 5 Conclusion

We propose ATENet, a novel end-to-end framework designed to enhance classification performance on irregular multivariate time series by learning their effective representations. In particular, we introduce ATE, which learns reference time points and generates representations at these reference points. This approach can successfully capture missingness patterns without information loss caused by disregarding uneven time intervals and without the need for an expensive tuning process to find optimal reference points. ATE also introduces temporal and intervariable consistency regularization terms, ensuring the enrichment of temporal information and efficient reflection of intervariable relationships. A series of experiments on irregular multivariate time-series classification demonstrated that ATENet outperformed the SOTA methods with high computational efficiency.

## Acknowledgments

This research was partially supported by the National Research Foundation of Korea (NRF) grant funded by the Ministry of Science and ICT (MSIT) (RS-2023-00208412); by the Korea Meteorological Administration Research and Development Program (KMI; RS-2025-02221093 and RS-2025-02219688); and by the U.S. National Institutes of Health (NIH) under Grant R01AG065330 (HD).

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

# A Overview of ATENet

In this study, we propose ATENet that sequentially combines an encoder $f$ configured by our encoding approach, ATE, with a classifier $g$ to enhance classification performance on irregular multivariate time series.

While ATENet might appear to incorporate familiar components such as attention-based interpolation and contrastive regularization, its key contribution lies in the effective integration and optimization of these components to address the unique challenges of irregular multivariate time-series classification.

- **Adaptive encoding process**: We propose a novel encoding framework that learns reference time points in an end-to-end supervised manner, eliminating the need for predefined anchors or handcrafted temporal discretization. Unlike prior approaches, our method directly optimizes the reference points with respect to both the task objective and the empirical distribution of observed time points. This design enables the model to align irregular sequences in a task-aware and data-adaptive temporal space, thereby improving both the expressiveness and generalization of the learned representations across varying sampling patterns.

- **Consistency regularization**: To further improve the quality and robustness of representations, ATENet incorporates two lightweight yet effective regularization techniques:

  - **Temporal consistency regularization**: This component enforces stability in the learned representations under random temporal masking, a perturbation strategy specialized for sparse or irregular time series. By promoting invariance to partially missing observations, it enhances generalization under temporal noise or missingness.

  - **Intervariable consistency regularization**: We design a novel contrastive objective that encourages structural consistency across variables by exploiting the outer product between input and representation spaces. This approach efficiently captures intervariable dependencies and serves as a lightweight alternative to complex graph-based or recurrent models, enhancing cross-variable coherence without incurring high computational overhead.

In Algorithm A1, we present a pseudo-code of our method to describe its overall learning procedure.

---

**Algorithm A1** Learning procedure of ATENet

---

**Input:** Set of $N$ labeled irregular multivariate time-series samples $\mathbb{D} = \{(\mathcal{X}_n, \boldsymbol{y}_n)\}_{n=1}^{N}$, set of time embedding functions $\Phi = \{\phi_1, \cdots, \phi_H\}$, and the number of learnable reference time points $K$
**Output:** Trained encoder $f$, classifier $g$, and reference time points $\boldsymbol{r} = \{r_1, \cdots, r_K\}$
 1: Initialize encoder $f$, classifier $g$, and reference points $\boldsymbol{r}$.
 2: **for** each epoch **do**
 3:     **for** $(\mathcal{X}_n, \boldsymbol{y}_n) \in \mathbb{D}$ **do**
 4:         *# Learnable Reference Time Points*
 5:         Obtain time embedding vectors for $\mathcal{X}_n$ and $\boldsymbol{r}$ using the $H$ time embedding functions in $\Phi$.
 6:         Generate $\boldsymbol{z}_n = \{z_{n1}, \cdots, z_{nK}\}$ at learnable reference time points $\boldsymbol{r}$ by Eqs. (3)-(5).
 7:         *# Temporal Consistency Regularization*
 8:         Randomly pick a masking ratio $\in [0.1, 0.9]$.
 9:         Generate a masked view $\boldsymbol{z}'_n = \{z'_{n1}, \cdots, z'_{nK}\}$ by Eqs. (4)-(6).
10:         Compute temporal consistency regularization term $\mathcal{L}_{TC_n}$ by Eqs. (7)-(9).
11:         *# Intervariable Consistency Regularization*
12:         Obtain two outer product matrices $\mathcal{P}_n$ and $\mathcal{Q}_n$ for $\mathcal{X}_n$ and $\boldsymbol{z}_n$, respectively, by Eq. (10).
13:         Compute intervariable consistency regularization term $\mathcal{L}_{VC_n}$ by Eq. (11).
14:         *# Classification Loss Function*
15:         $\hat{\boldsymbol{y}}_n \leftarrow g(\boldsymbol{z}_n)$
16:         Compute classification loss $\mathcal{L}_{CL_n}$ by Eq. (12).
17:     **end for**
18:     *# Optimization*
19:     Update $f, g$ and $\boldsymbol{r}$ by Eq. (13).
20: **end for**

---

| Dataset | Number of samples | Number of variables | Number of observed points | Number of classes | Sparsity ratio (%) |
|---------|-------------------|---------------------|---------------------------|-------------------|--------------------|
| P12-M   | 11,988            | 36                  | 215                       | 2                 | 88.4               |
| P12-L   | 11,988            | 36                  | 215                       | 2                 | 88.4               |
| P19     | 38,803            | 34                  | 60                        | 2                 | 94.9               |
| PAM     | 5,333             | 17                  | 600                       | 8                 | 60.0               |

Table A1: Dataset statistics of three datasets. *Sparsity ratio* is the ratio between the number of missing observations and that of all possible observations when fully observed.

## B  Complexity Analysis

Here, let $N$, $V$, $C$, and $T$ be the number of instances, variables, classes, and the maximum sequence length in an irregular multivariate time-series dataset $\mathbb{D}$, respectively. In addition, let $H$ be the number of embedding functions (attention heads) in the encoder $f$ and $J$ be the maximum hidden dimension in the classifier $g$, respectively.

ATE consists of the procedures for interpolation on learnable reference points based on the multi-time attention mechanism (MTA) [30] and computing two consistency regularization terms.

- *Interpolation on learnable reference points* leverages the MTA that has time complexities for computing query, key, and value matrices for $H$ attention heads ($O(NV^2K + NV^2T)$), calculating the scaled dot product attention for $H$ heads ($O(NVKT)$), and concatenating the results from $H$ heads with a linear transform ($O(NV^2K)$). Thus, the time complexity of this procedure is summarized as $O(NV(VK + VT + KT))$.
- *Temporal consistency regularization* proceeds the random masking to inputs ($O(NVT)$) and additional MTA for the masked inputs ($O(NV(VK+VT+KT))$); hence, its time complexity is also dominated by $O(NV(VK + VT + KT))$.
- *Intervariable consistency regularization* has time complexities for computing the outer product of variables and that of representations ($O(NV(VT + VK))$).

Therefore, the time complexity per epoch of ATE is dominated by $O(NV(VK+VT+KT))$. In general, $K \leq T$; hence, the time complexity can be reduced to $O(NVT(V+K))$. While the attention mechanism can incur a relatively large computational cost, our approach introduces minimal computational burden for reflecting intervariable relations, compared to existing methods that account for intervariable dependencies in multivariate irregular time series by leveraging graph neural networks.

ATENet consists of the encoder with the time complexity of $O(NVT(V + K))$. The classifier in ATENet is constructed by a gated recurrent unit (GRU) followed by two fully connected layers with batch normalization and a GeLU activation function; hence, it requires the additional time complexity of $O(NJ(VK + KJ + C))$. Thus, ATENet has the time complexity per epoch of $O(NVT(V + K) + NJ(VK + KJ + C))$. When $C$ is smaller than $J$, the time complexity per epoch can be reduced to $O(NVT(V + K) + NJ(VK + KJ))$. In Figure 5 of the main body, we compare our method to existing methods in terms of the number of parameters and processing time per epoch.

## C  Detailed Description on Datasets

To evaluate the proposed method, ATENet, we employed three irregular multivariate time-series datasets as follows:

- *P12* (PhysioNet Mortality Prediction Challenge 2012) [7] is one of the popular healthcare datasets recorded by 36 sensors for 11,988 patients, after dropping 12 inappropriate samples [9], during their first 48-hour stay in the intensive care unit. This dataset has two predictive labels: in-hospital mortality (*P12-M*) and hospitalization length (*P12-L*). A positive class of the P12-M dataset indicates in-hospital death, and this dataset has 13.8% positive samples. For the P12-L dataset, each patient is assigned a binary class label indicating the length of hospitalization. While a positive class indicates a stay longer than three days, a negative class indicates a stay of three days or less. This dataset has 93% positive samples, thereby being highly imbalanced.
- *P19* (PhysioNet Sepsis Early Prediction Challenge 2019) [27] is another popular healthcare dataset containing 40,336 patients monitored by 34 irregularly sampled sensors, including

eight vital signs and 26 laboratory values. We remove the samples with extremely short or long time series, thereby remaining 38,803 patients with more than one and less than 60 observations. Each patient has a binary class label that indicates the occurrence of sepsis within the next six hours. This dataset is highly imbalanced due to about 96% negative samples.

- *PAM* (PAMAP2 Physical Activity Monitoring) [25] PAM has 5,333 samples with 600 continuous observations measured by 17 sensors after modification following Zhang et al. [47]. This dataset has eight human activities of daily life. To make the samples irregular, we randomly removed 60% of the observations. Each sample is annotated with one of eight activities of daily living. The number of samples is balanced across all eight activities.

The characteristics of each dataset are given in Table A1. To handle missing values in irregular time series, we first set the missing values to zero. The input was then configured by concatenating the observations with a binary mask indicator, which was set to 1 when an observation existed and 0 otherwise. Following Zhang et al. [47], for the P12-M, P12-L, and P19 datasets, we applied batch minority class upsampling. We also excluded static information (e.g., age, gender, height, and weight) from model training to focus on each method's ability to capture temporal patterns within irregular time series. All datasets were split into training (80%), validation (10%), and test (10%) sets.

## D   Implementation Details

In ATENet, the encoder $f$ maps an irregular multivariate time series $\mathcal{X} = (t, x)$ with $V$ variables to $V$-dimensional time embeddings at $K$ reference time points. The encoder $f$ consists of multi-head attention with $H$ embedding functions (attention heads), each deriving a time embedding vector of size $L$ for each observed time point. In this study, both $K$ and $L$ were set to 128. For the P12-M, P12-L, P19, and PAM datasets, we set $H$ to 2, 4, 2, and 1, respectively, while using *learnable*, *learnable*, *sinusoidal*, and *learnable* embedding functions in that order. The subsequent classifier $g$ is constructed by a GRU followed by two fully connected layers, where the first layer includes batch normalization and a GeLU activation function, and the second one uses a softmax as an activation function. The GRU dimension was set to 32, and the two fully connected layers had dimensions of 32 and $C$, respectively, where $C$ is the number of classes. Additionally, the scaling parameter $\epsilon$ in Eq. (3) and the smoothing parameter $\eta$ in Eq. (12) of the main text were set to 128 and 0.1, respectively.

For model training, we set the batch size $B$ to 128 and ran the model for 20 epochs. For the P12-M dataset, we used the Adam optimizer with an initial learning rate of 0.0001 and assigned weights of 0.01 and 0.1 to the temporal and intervariable consistency regularization terms, $\alpha$ and $\beta$, respectively. In the P12-L dataset, we used the Adam optimizer with an initial learning rate of 0.001 and set both $\alpha$ and $\beta$ to 0.01. In the P19 dataset, we also employed the Adam optimizer, but with an initial learning rate of 0.01, and adjusted $\alpha$ and $\beta$ to 0.01 and 1, respectively. For the PAM dataset, we maintained a learning rate of 0.01, while setting $\alpha$ to 0.1 and $\beta$ to 0.01. Note that all hyperparameters were selected based on performance on the validation set, and the final results were obtained from the model that achieved the best validation performance. The impact of each hyperparameter is further explored in Appendix I.

We repeated each experiment five times and reported the averages and standard deviations. All experiments were executed on a PyTorch platform using an Intel Core i9-10900X at 3.70 GHz CPU, 256 GB RAM, and an NVIDIA GeForce RTX 4090 24 GB GPU.

## E   Statistical Tests

To confirm the significance of the differences in classification performance in Table 1 of the main text, we conducted statistical tests. Specifically, we adopted a paired t-test to compare the proposed method to each baseline on each dataset. The following markers indicate the results of the significance tests:

- ✔ indicates that ATENet performed significantly better than the baseline with a smaller p-value than 0.05.
- − indicates no significant difference between the methods being compared.
- ✗ indicates the baseline performed significantly better than our approach.

| Metric | Dataset | mTAND | DGM² | GRU-D | MTGNN | Transformer | Trans-mean | SeFT | Raindrop | Warpformer | MTSFormer |
|---|---|---|---|---|---|---|---|---|---|---|---|
| AUROC | P12-M | – | ✔ | ✔ | ✔ | ✔ | ✔ | ✔ | ✔ | ✔ | ✔ |
| | P12-L | ✔ | ✔ | ✔ | ✔ | ✔ | ✔ | ✔ | ✔ | – | – |
| | P19 | ✔ | – | ✗ | – | ✔ | ✔ | ✔ | – | – | ✗ |
| | PAM | ✔ | ✔ | ✔ | ✔ | ✔ | ✔ | ✔ | ✔ | ✔ | ✔ |
| AUPRC | P12-M | – | ✔ | ✔ | ✔ | ✔ | ✔ | ✔ | ✔ | ✔ | ✔ |
| | P12-L | ✔ | – | ✔ | ✔ | ✔ | ✔ | ✔ | – | – | – |
| | P19 | ✔ | ✔ | ✗ | – | ✔ | ✔ | ✔ | ✗ | – | ✗ |
| | PAM | ✔ | ✔ | ✔ | ✔ | ✔ | ✔ | ✔ | ✔ | ✔ | ✔ |

Table A2: Statistical significance results comparing ATENet with each baseline on each dataset

As shown in Table A2, the proposed method, ATENet, performed significantly better than the baselines in most cases.

## F  Limitations

Here, we discuss some limitations of our work and suggest potential directions for future research.

### F.1  Marginal Performance Gains and Trade-offs

As shown in Tables 1 and A2, our method did not outperform some baseline methods, particularly GRU-D, Raindrop, and MTSFormer, on the P19 dataset. While the learnable reference time points are designed to adapt to irregular patterns, the P19 dataset may exhibit extreme variability and noise in sampling times and values, making it challenging for any method based on reference time points to capture nuanced patterns effectively and generalize the reference time points for all data [6, 27]. For example, as shown in Table 1 of the main text, our method and mTAND, which leverage reference time points, showed relatively low classification performance on the P19 dataset compared to the other datasets. Therefore, in future work, we expect to enhance classification performance on such challenging datasets by incorporating additional techniques, such as Kalman filter or wavelet denoising, to mitigate the effects of extreme variability and noise in sampling.

Furthermore, our key contribution lies in the encoder, which transforms irregular multivariate time series into fixed-length representations at learnable reference points. To demonstrate its effectiveness without relying on complex backbones, we deliberately used a simple classifier. However, more complex classifiers may improve performance by capturing richer class-specific patterns. In preliminary experiments on the P12-M dataset, replacing the simple decoder with a transformer and a temporal convolutional network yielded slight AUROC improvements of 0.13% and 0.50%, respectively, but with 1.08× and 2.29× increases in processing time per epoch, respectively. These marginal gains suggest that classifier complexity may trade off with the overall efficiency of our method. Designing classifiers that balance accuracy and efficiency is a promising direction for future work.

### F.2  Robustness to Start-Time Mismatch

First, start-time mismatch (or misalignment) is not inherently problematic in modern sequence modeling approaches such as Transformers or recurrent models [24, 29, 20, 17, 35, 2]. These models do not rely on absolute timestamps but instead learn representations based on the relative positions or temporal dynamics within each sequence. Consequently, the fact that sequences start at different times does not invalidate the model's ability to learn meaningful patterns, nor does it by itself constitute a distribution shift.

Second, in many real-world time-series domains, start times are often not entirely random. Instead, they tend to follow domain-specific routines or event triggers. For example, patient monitoring typically begins at symptom onset or ICU admission in clinical data, and measurements in industrial settings may align with batch starts or fault occurrences. Thus, even without explicit synchronization, there is often implicit regularity that the model can exploit. ATENet is designed to capture such latent patterns by learning reference points and generating representations aligned to them. Moreover, even in scenarios where start times are highly variable or nearly random, ATENet is still likely to work well by learning uniformly distributed reference points that provide broad temporal coverage without depending on strict alignment.

### F.3 Generalization under Temporal Distribution Shifts

Regarding generalization to datasets with different temporal distributions (i.e., distribution shift), our experiments were conducted under within-dataset settings. However, shifts in temporal dynamics, such as variations in event density or measurement frequency between training and deployment environments, may affect the effectiveness of the learned reference points. In such cases, the learned reference points may fail to align with informative regions in the target data, potentially degrading interpolation quality and downstream performance. We identify this limitation as an important direction for future work.

### F.4 Robustness under Extremely Sparse Observation Scenarios

Similar to other attention-based methods, ATENet may underperform in extremely sparse observation scenarios due to the fundamental lack of information. However, ATENet is designed to be more robust to such sparsity than conventional approaches by:

- Learning reference points that adapt to data-driven observation patterns, especially around informative regions;

- Using attention-based interpolation that effectively aggregates information from both nearby and even distant observations;

- Introducing temporal consistency regularization based on random masking, which enhances representation quality even when data is partially missing.

As shown in Appendix H, ATENet demonstrates strong robustness under high levels of missingness. Specifically, the model maintained competitive performance when up to 50% of the observations were randomly dropped, indicating that our method remains effective as long as a minimal level of temporal coverage is preserved.

Nonetheless, in extremely sparse settings where few or no observations exist near any reference point, performance can degrade. We will address this limitation in future work that is robust in extremely sparse scenarios.

## G Complete Results

Here, we present the complete results of the experiments, which demonstrate robustness to missing variables and the effectiveness of temporal and intervariable consistency regularization terms.

### G.1 Robustness to Missing Variables

ATENet can mitigate a drastic performance drop, when some variables are removed, by introducing intervariable consistency regularization to capture structural relations between variables in irregular multivariate time series. In Section 4.2 of the main text, we investigated the robustness of ATENet by selecting a subset of variables and hiding their observations in the test set. Following Zhang et al. [47], we considered two scenarios: *leave-fixed-sensors-out* and *leave-random-sensors-out*.

In Tables A3 and A4, we provide the complete results for the classification performance of ATENet and that of the baseline methods when eliminating variables by various ratios $\in [0.1, 0.5]$ for both scenarios, respectively. In most cases, the proposed method performed more robustly than the baseline methods by achieving a higher classification performance. In contrast, although MTGNN, Raindrop, Warpformer, and MTSFormer consider intervariable relationships, their classification performance significantly declined in most datasets. For example, in the PAM dataset under the *leave-fixed-sensors-out* scenario, MTGNN exhibited a maximum performance drop of 30% in AUROC and 60% in AUPRC. Raindrop showed a maximum performance drop of 40% in AUROC and 70% in AUPRC. Both Warpformer and MTSFormer showed about 50% and 80% performance drop in AUROC and AUPRC, respectively.

These results demonstrate the robustness of ATENet against missing variables by successfully capturing intervariable relationships in irregular multivariate time series.

| Metric | Dataset | Drop Ratio | mTAND | DGM² | GRU-D | MTGNN | Transformer | Trans-mean | SeFT | Raindrop | Warpformer | MTSFormer | ATENet |
|---|---|---|---|---|---|---|---|---|---|---|---|---|---|
| AUROC | P12-M | - | 84.18±1.20 | 71.08±2.30 | 48.62±2.41 | 61.59±5.79 | 82.92±0.72 | 83.38±0.56 | 68.05±1.49 | 81.19±1.76 | 79.35±1.65 | 84.11±0.71 | 85.54±1.26 |
| | | 0.1 | 83.41±1.10 | 67.42±2.71 | 48.80±2.58 | 58.13±4.03 | 74.58±1.42 | 74.23±1.00 | 67.83±1.44 | 58.33±8.45 | 77.56±1.23 | 77.09±2.06 | 85.12±1.24 |
| | | 0.2 | 73.41±2.81 | 66.64±2.75 | 49.80±1.18 | 57.87±3.50 | 71.67±1.87 | 70.08±1.86 | 67.89±1.89 | 64.61±1.41 | 52.83±18.17 | 74.71±2.31 | 75.94±2.59 |
| | | 0.3 | 69.64±5.02 | 64.72±1.82 | 49.05±1.26 | 56.14±3.52 | 64.81±3.30 | 64.44±3.36 | 67.70±1.85 | 62.28±4.02 | 51.11±13.05 | 65.16±2.56 | 69.32±2.71 |
| | | 0.4 | 69.88±3.85 | 62.70±0.74 | 47.14±1.54 | 54.47±5.72 | 60.65±2.35 | 60.67±2.70 | 67.53±1.89 | 60.20±1.79 | 50.33±7.65 | 58.78±3.52 | 69.04±2.86 |
| | | 0.5 | 65.69±3.78 | 63.25±1.94 | 47.17±1.61 | 54.87±4.51 | 57.99±2.51 | 57.99±2.51 | 65.05±1.86 | 60.52±3.31 | 49.92±0.16 | 55.44±1.93 | 62.32±3.06 |
| | P12-L | - | 49.60±3.16 | 69.46±1.47 | 49.82±3.85 | 68.36±6.09 | 59.05±1.81 | 61.64±1.54 | 64.70±2.01 | 70.40±1.60 | 74.57±2.28 | 75.17±1.09 | 79.64±2.24 |
| | | 0.1 | 49.63±3.15 | 69.18±1.39 | 49.83±3.85 | 67.69±5.76 | 58.98±1.80 | 60.11±1.69 | 64.55±2.08 | 66.90±1.22 | 72.71±2.79 | 72.94±1.72 | 79.63±2.38 |
| | | 0.2 | 55.32±4.13 | 69.16±1.34 | 51.73±4.16 | 67.63±5.77 | 58.91±1.81 | 59.86±1.60 | 64.32±2.06 | 66.80±1.45 | 55.56±14.43 | 72.58±1.76 | 75.81±2.47 |
| | | 0.3 | 55.06±2.24 | 68.11±1.73 | 51.34±1.56 | 68.22±4.00 | 56.72±2.65 | 57.77±2.30 | 64.24±2.05 | 66.87±2.09 | 55.03±7.72 | 68.58±2.57 | 74.79±2.51 |
| | | 0.4 | 53.42±2.50 | 68.02±2.08 | 53.24±1.59 | 67.96±4.85 | 56.17±2.77 | 57.43±2.36 | 64.29±1.98 | 67.60±2.92 | 53.93±8.34 | 67.70±2.03 | 74.56±2.61 |
| | | 0.5 | 54.56±2.50 | 67.81±2.13 | 52.59±3.52 | 67.69±4.99 | 53.35±2.91 | 53.35±2.91 | 63.37±2.07 | 66.70±1.84 | 49.85±0.26 | 66.82±2.21 | 73.38±2.55 |
| | P19 | - | 80.00±1.23 | 81.96±2.05 | 87.16±1.34 | 85.07±3.54 | 77.56±3.06 | 78.57±3.02 | 77.89±2.62 | 85.93±2.24 | 85.41±2.39 | 85.40±1.66 | 84.02±1.38 |
| | | 0.1 | 78.79±1.16 | 34.78±27.69 | 86.26±1.90 | 81.77±3.50 | 75.54±3.53 | 75.82±3.40 | 77.00±1.99 | 82.86±1.67 | 72.54±3.44 | 84.67±1.70 | 82.47±1.45 |
| | | 0.2 | 78.34±1.34 | 34.28±27.52 | 85.10±0.89 | 81.79±3.21 | 74.98±3.52 | 75.20±3.43 | 74.74±1.10 | 80.52±1.73 | 45.08±9.12 | 84.68±2.30 | 77.88±1.99 |
| | | 0.3 | 79.58±0.73 | 34.11±27.45 | 84.64±1.98 | 82.01±3.23 | 74.94±3.53 | 75.13±3.53 | 68.01±1.68 | 80.58±2.03 | 48.00±5.63 | 84.18±2.60 | 76.76±1.85 |
| | | 0.4 | 77.47±2.42 | 34.12±27.41 | 81.67±1.36 | 81.82±3.08 | 74.75±3.49 | 74.86±3.39 | 49.33±28.51 | 60.20±1.79 | 51.24±4.07 | 79.22±1.58 | 76.38±1.78 |
| | | 0.5 | 76.30±2.75 | 34.10±27.29 | 79.62±1.65 | 81.29±3.50 | 74.67±3.50 | 74.67±3.50 | 47.35±27.37 | 80.05±2.39 | 50.04±0.11 | 75.86±1.39 | 74.55±2.12 |
| | PAM | - | 92.21±0.70 | 96.87±0.50 | 91.72±0.59 | 96.95±0.32 | 96.61±1.27 | 97.64±0.25 | 74.46±6.70 | 98.73±0.25 | 97.94±0.45 | 98.39±0.28 | 99.18±0.15 |
| | | 0.1 | 89.62±1.17 | 95.74±1.08 | 81.33±2.60 | 95.92±0.98 | 94.76±1.61 | 93.74±1.34 | 68.16±1.97 | 71.89±13.61 | 96.16±1.02 | 96.23±0.72 | 99.15±0.16 |
| | | 0.2 | 78.05±2.66 | 90.84±2.14 | 78.85±3.00 | 89.85±1.03 | 85.48±1.23 | 85.07±1.45 | 66.71±1.75 | 63.87±7.59 | 49.27±4.42 | 81.28±4.46 | 95.55±1.36 |
| | | 0.3 | 71.11±4.19 | 88.04±2.40 | 73.11±5.54 | 86.40±2.15 | 82.33±1.94 | 82.83±1.84 | 68.64±1.47 | 60.43±15.36 | 47.56±3.61 | 77.06±3.40 | 94.53±1.60 |
| | | 0.4 | 69.16±5.60 | 86.86±1.70 | 61.55±2.33 | 82.53±3.57 | 81.51±1.83 | 82.98±1.91 | 63.47±2.38 | 60.79±9.98 | 53.09±11.73 | 74.65±3.38 | 87.94±3.38 |
| | | 0.5 | 68.45±5.58 | 69.12±2.67 | 60.41±1.64 | 68.53±1.47 | 50.11±0.53 | 50.11±0.53 | 58.85±1.27 | 59.85±3.91 | 50.09±0.30 | 50.36±0.38 | 87.76±3.33 |
| AUPRC | P12-M | - | 52.89±2.27 | 29.99±2.24 | 14.83±1.55 | 24.25±5.49 | 46.35±2.81 | 48.54±2.24 | 24.43±3.10 | 42.14±3.32 | 41.98±1.30 | 48.53±2.55 | 53.31±2.02 |
| | | 0.1 | 50.84±2.13 | 27.29±2.47 | 13.46±1.03 | 21.38±3.00 | 36.30±2.75 | 36.10±3.45 | 24.41±3.10 | 20.26±6.18 | 38.48±2.81 | 39.55±2.35 | 51.59±2.48 |
| | | 0.2 | 34.48±6.24 | 26.91±2.61 | 15.11±1.24 | 21.43±2.83 | 33.98±1.90 | 33.16±1.31 | 24.48±2.44 | 22.89±3.49 | 20.68±9.75 | 36.80±2.55 | 38.57±3.96 |
| | | 0.3 | 26.48±5.07 | 25.53±2.56 | 14.64±0.67 | 19.88±2.18 | 24.89±3.17 | 24.64±3.32 | 24.59±2.64 | 21.56±3.93 | 17.69±6.17 | 24.95±1.94 | 29.31±2.38 |
| | | 0.4 | 27.91±4.45 | 24.01±2.20 | 13.99±1.43 | 18.28±3.66 | 20.65±3.03 | 20.90±2.85 | 24.57±2.82 | 20.79±2.82 | 15.84±2.68 | 20.62±3.82 | 29.12±2.56 |
| | | 0.5 | 23.21±1.99 | 22.53±2.24 | 13.25±0.98 | 18.68±2.92 | 18.57±3.29 | 18.57±3.29 | 22.60±2.85 | 20.43±4.02 | 14.44±1.32 | 17.48±3.30 | 22.40±1.54 |
| | P12-L | - | 92.42±1.09 | 96.42±0.41 | 93.41±0.93 | 96.41±1.08 | 94.16±0.99 | 94.65±0.80 | 95.28±0.24 | 96.57±0.51 | 96.99±0.34 | 97.43±0.28 | 97.70±0.38 |
| | | 0.1 | 92.43±1.07 | 96.34±0.43 | 93.41±0.90 | 96.28±1.02 | 94.17±0.98 | 94.24±1.05 | 95.28±0.25 | 96.01±0.37 | 96.84±0.65 | 97.11±0.45 | 97.74±0.41 |
| | | 0.2 | 94.14±1.25 | 96.33±0.44 | 93.46±1.07 | 96.27±1.02 | 94.28±0.99 | 94.26±1.00 | 95.23±0.24 | 96.08±0.39 | 93.30±3.53 | 97.05±0.46 | 97.04±0.53 |
| | | 0.3 | 94.28±0.39 | 96.27±0.59 | 92.83±0.88 | 96.49±0.89 | 94.09±1.12 | 94.06±1.10 | 95.24±0.30 | 96.17±0.51 | 93.33±2.00 | 96.12±0.69 | 96.81±0.62 |
| | | 0.4 | 93.51±0.43 | 96.27±0.59 | 93.52±0.46 | 96.48±0.93 | 93.97±1.11 | 94.03±1.07 | 95.26±0.29 | 96.15±0.63 | 93.35±1.79 | 95.71±0.60 | 96.85±0.63 |
| | | 0.5 | 94.12±1.05 | 96.28±0.54 | 93.33±0.29 | 96.39±0.96 | 93.47±1.18 | 93.47±1.18 | 95.17±0.32 | 95.94±0.58 | 93.09±0.55 | 95.50±0.67 | 96.91±0.52 |
| | P19 | - | 31.24±4.15 | 31.12±5.25 | 47.37±2.97 | 41.13±8.01 | 29.60±6.26 | 28.05±6.23 | 30.34±1.80 | 50.63±3.32 | 41.12±3.30 | 57.96±4.10 | 41.16±3.02 |
| | | 0.1 | 28.85±3.02 | 9.62±12.49 | 38.93±5.54 | 39.57±6.96 | 36.38±5.69 | 26.82±5.73 | 25.97±1.74 | 46.08±4.96 | 18.58±2.15 | 49.36±3.00 | 36.42±2.75 |
| | | 0.2 | 29.09±3.17 | 10.08±13.50 | 28.09±3.66 | 40.51±6.03 | 38.16±6.29 | 30.40±7.19 | 15.77±0.68 | 44.29±4.29 | 5.18±2.58 | 48.14±4.62 | 26.14±5.83 |
| | | 0.3 | 31.81±3.06 | 10.34±13.71 | 26.99±4.19 | 40.64±5.61 | 39.19±5.88 | 32.57±6.84 | 13.29±1.01 | 44.18±4.47 | 4.79±0.66 | 47.53±4.72 | 25.11±5.64 |
| | | 0.4 | 27.37±2.97 | 10.26±13.91 | 20.58±2.31 | 40.75±5.99 | 40.11±5.41 | 34.70±6.05 | 9.37±5.52 | 20.79±2.82 | 4.65±0.58 | 43.22±4.31 | 23.72±5.43 |
| | | 0.5 | 24.54±3.14 | 9.94±13.32 | 21.81±3.42 | 40.19±6.12 | 40.26±5.04 | 40.26±5.04 | 7.29±4.36 | 43.80±4.38 | 4.35±0.25 | 41.39±3.74 | 20.45±6.16 |
| | PAM | - | 74.95±2.68 | 88.28±1.28 | 75.78±2.02 | 88.85±2.00 | 86.73±4.21 | 91.50±0.61 | 36.43±12.23 | 95.48±0.91 | 92.75±1.43 | 94.21±0.71 | 97.61±0.26 |
| | | 0.1 | 67.36±4.45 | 84.01±3.02 | 56.01±4.78 | 83.53±5.56 | 77.00±5.41 | 72.29±5.03 | 26.75±2.54 | 37.51±25.55 | 86.42±3.32 | 84.32±2.94 | 97.42±0.20 |
| | | 0.2 | 42.36±5.01 | 66.01±4.97 | 50.66±5.61 | 63.05±3.46 | 50.53±2.57 | 51.47±3.34 | 26.51±2.61 | 27.07±5.29 | 15.67±3.64 | 52.55±4.11 | 89.14±2.14 |
| | | 0.3 | 30.72±5.62 | 57.26±4.53 | 41.54±10.12 | 52.11±4.92 | 43.28±2.18 | 45.06±2.61 | 27.11±1.46 | 24.77±9.91 | 15.46±1.48 | 39.02±2.61 | 84.78±3.09 |
| | | 0.4 | 29.18±6.40 | 54.30±3.81 | 24.54±5.33 | 47.25±7.31 | 40.75±1.37 | 45.58±2.14 | 24.07±1.93 | 24.97±7.64 | 18.20±6.21 | 36.78±2.02 | 63.76±5.85 |
| | | 0.5 | 27.65±5.85 | 30.79±3.55 | 22.14±3.24 | 29.44±1.73 | 12.63±0.17 | 12.63±0.17 | 20.17±2.33 | 22.12±4.29 | 12.55±0.05 | 12.76±0.13 | 62.31±6.22 |

Table A3: Classification performance of ATENet and baselines when dropping variables with various ratios $\in [0.1, 0.5]$ in the *leave-fixed-sensors-out* scenario. *Drop Ratio* denotes the ratio of missing variables.

## G.2   Effects of Temporal and Intervariable Consistency Regularization

To validate the influence of the temporal and intervariable consistency regularization terms, we compared the classification performance of ATENet to that of ablation models, ATENet *w/o* $\mathcal{L}_{TC}$, ATENet *w/o* $\mathcal{L}_{TCP}$, ATENet *w/o* $\mathcal{L}_{TCI}$ and ATENet *w/o* $\mathcal{L}_{VC}$ in Section 4.3 of the main text. Table A5 presents the complete results of these ablation studies.

As a result, ATENet outperformed the ablation models in most cases, demonstrating the effectiveness of both temporal and intervariable consistency regularization terms in enhancing classification performance. Specifically, removing temporal consistency (ATENet *w/o* $\mathcal{L}_{TC}$) exhibited an average drop in AUPRC of 1.28%, whereas removing intervariable consistency (ATENet *w/o* $\mathcal{L}_{VC}$) caused a larger average drop of 3.33%. These findings indicate that both components are beneficial, with intervariable consistency showing a more substantial effect in our experimental settings.

## G.3   Futher Analysis

### G.3.1   Empirical Trade-offs Between Flexibility and Stability

We evaluated a variant that computes reference points individually for each sample. Although this approach offers better flexibility, it can be more sensitive to noise and irregular sampling, often resulting in unstable training and degraded generalization performance.

As shown in the Table A6, the globally shared design (ours) showed comparable or slightly better performance across most datasets. Notably, the fully individualized variant showed a substantial drop in performance on the P19 dataset, which may contain highly variable and noisy sequences (see Appendix F). This highlights that globally shared reference points, combined with sample-specific attention-based interpolation, offer better robustness under challenging conditions.

| Metric | Dataset | Drop Ratio | mTAND | DGM² | GRU-D | MTGNN | Transformer | Trans-mean | SeFT | Raindrop | Warpformer | MTSFormer | ATENet |
|---|---|---|---|---|---|---|---|---|---|---|---|---|---|
| AUROC | P12-M | - | 84.18±1.20 | 71.08±2.30 | 48.62±2.41 | 61.59±5.79 | 82.92±0.72 | 83.38±0.56 | 68.05±1.49 | 81.19±1.76 | 79.35±1.65 | 84.11±0.71 | 85.54±1.26 |
| | | 0.1 | 76.75±1.53 | 69.55±1.94 | 49.69±1.42 | 61.25±5.48 | 81.38±0.61 | 81.18±0.88 | 64.04±1.84 | 58.30±7.80 | 80.42±1.20 | 82.40±1.15 | 84.25±1.85 |
| | | 0.2 | 73.15±2.15 | 68.65±1.51 | 49.83±1.25 | 60.82±4.53 | 80.43±1.15 | 80.24±1.84 | 61.15±2.05 | 63.48±5.45 | 54.59±22.51 | 81.68±1.17 | 82.06±1.50 |
| | | 0.3 | 69.11±1.77 | 66.88±0.88 | 48.90±2.30 | 57.77±3.93 | 77.23±0.70 | 77.17±0.91 | 61.53±2.11 | 57.93±6.10 | 51.40±19.43 | 78.29±0.69 | 79.46±1.39 |
| | | 0.4 | 66.32±1.43 | 64.40±1.24 | 47.65±0.69 | 57.33±3.74 | 76.53±1.95 | 77.09±1.14 | 59.64±2.31 | 59.86±5.80 | 46.85±17.77 | 76.52±2.62 | 78.86±1.53 |
| | | 0.5 | 63.25±2.07 | 64.09±1.24 | 49.54±3.00 | 57.42±2.55 | 74.28±0.63 | 74.67±0.86 | 57.87±3.35 | 62.42±4.79 | 49.39±17.51 | 73.78±1.34 | 76.48±2.22 |
| | P12-L | - | 49.60±3.16 | 69.46±1.47 | 49.82±3.85 | 68.36±6.09 | 59.05±1.81 | 61.64±1.54 | 64.70±2.01 | 70.40±1.60 | 74.57±2.28 | 75.17±1.09 | 79.64±2.24 |
| | | 0.1 | 54.33±5.35 | 69.14±1.87 | 53.68±3.61 | 67.55±6.67 | 58.88±2.12 | 60.29±1.98 | 63.74±2.43 | 66.92±1.20 | 73.94±2.63 | 74.33±1.51 | 78.48±1.78 |
| | | 0.2 | 55.62±3.93 | 68.64±1.24 | 52.50±3.30 | 66.65±5.90 | 58.91±2.52 | 60.21±2.15 | 62.54±2.88 | 66.71±1.52 | 54.66±18.23 | 72.69±2.18 | 78.79±2.11 |
| | | 0.3 | 56.18±3.78 | 69.13±1.28 | 52.30±2.52 | 65.47±6.23 | 57.64±2.11 | 58.99±1.97 | 63.47±1.24 | 66.57±1.60 | 54.15±13.82 | 70.87±2.15 | 76.74±2.13 |
| | | 0.4 | 54.99±3.02 | 66.80±1.64 | 53.61±3.95 | 63.59±6.06 | 57.52±2.68 | 58.36±2.45 | 62.81±2.35 | 67.25±2.29 | 48.78±16.58 | 70.82±0.85 | 76.10±1.98 |
| | | 0.5 | 52.69±3.41 | 66.70±3.35 | 51.83±3.04 | 63.00±5.08 | 56.95±2.29 | 58.18±1.90 | 61.89±3.44 | 61.25±2.29 | 45.43±13.54 | 69.77±1.22 | 75.78±1.85 |
| | P19 | - | 80.00±1.23 | 81.96±2.05 | 87.16±1.34 | 85.07±3.54 | 77.56±3.06 | 78.57±3.02 | 77.89±2.62 | 85.93±2.24 | 85.41±2.39 | 88.96±2.01 | 84.02±1.38 |
| | | 0.1 | 77.48±1.87 | 37.19±27.42 | 84.49±0.90 | 81.35±2.93 | 77.15±3.20 | 77.31±2.99 | 77.06±2.68 | 81.51±2.77 | 83.36±2.44 | 87.18±1.85 | 84.04±1.56 |
| | | 0.2 | 76.75±1.47 | 38.07±27.73 | 82.04±2.19 | 79.97±2.70 | 77.02±3.09 | 77.11±2.57 | 75.86±2.71 | 77.89±2.46 | 46.05±19.12 | 85.83±2.21 | 83.18±2.30 |
| | | 0.3 | 76.68±1.19 | 43.78±16.00 | 79.95±1.28 | 77.35±3.94 | 76.53±2.85 | 76.62±2.66 | 75.01±3.05 | 79.65±3.10 | 43.69±19.59 | 84.78±1.39 | 82.14±2.86 |
| | | 0.4 | 72.09±1.80 | 43.76±15.36 | 78.88±0.75 | 74.36±3.05 | 76.47±3.49 | 76.52±3.28 | 73.50±2.79 | 78.83±2.19 | 48.39±14.38 | 83.53±3.19 | 81.48±4.10 |
| | | 0.5 | 73.34±1.81 | 44.84±14.47 | 77.04±1.35 | 72.20±3.23 | 76.07±3.67 | 76.23±3.59 | 73.04±3.07 | 77.67±2.89 | 52.73±6.53 | 82.59±1.39 | 80.55±5.21 |
| | PAM | - | 92.21±0.70 | 96.87±0.50 | 91.72±0.59 | 96.95±0.32 | 96.61±1.27 | 97.64±0.25 | 74.46±6.70 | 98.73±0.25 | 97.94±0.45 | 98.39±0.28 | 99.18±0.15 |
| | | 0.1 | 83.24±0.99 | 95.53±0.66 | 86.28±1.06 | 95.22±0.40 | 95.51±1.06 | 94.53±1.07 | 68.13±1.15 | 72.96±23.13 | 96.80±0.51 | 97.43±0.34 | 98.39±0.13 |
| | | 0.2 | 77.94±1.21 | 92.29±0.34 | 83.81±0.55 | 92.47±0.81 | 93.42±1.16 | 93.01±0.93 | 65.75±0.98 | 64.10±4.99 | 51.03±7.47 | 94.96±0.68 | 97.40±0.60 |
| | | 0.3 | 71.40±1.50 | 90.14±1.07 | 75.25±1.36 | 90.14±0.60 | 91.92±0.98 | 92.01±0.90 | 63.71±1.56 | 64.21±7.49 | 46.09±3.27 | 92.78±1.05 | 95.23±0.74 |
| | | 0.4 | 66.96±1.11 | 86.19±0.96 | 68.73±1.68 | 86.39±1.59 | 89.43±0.43 | 89.78±0.55 | 31.40±0.83 | 60.23±3.97 | 49.96±7.27 | 89.50±1.65 | 92.84±1.05 |
| | | 0.5 | 64.51±1.83 | 83.16±0.77 | 65.62±1.65 | 82.62±0.57 | 86.94±0.46 | 87.41±0.47 | 60.55±1.36 | 60.24±2.56 | 50.54±7.35 | 85.95±2.19 | 90.98±1.31 |
| AUPRC | P12-M | - | 52.89±2.27 | 29.99±2.24 | 14.83±1.55 | 24.25±5.49 | 46.35±2.81 | 48.54±2.24 | 24.43±3.10 | 42.14±3.32 | 41.98±1.30 | 48.53±2.55 | 53.31±2.02 |
| | | 0.1 | 32.94±1.69 | 28.67±2.30 | 14.52±0.42 | 23.10±4.71 | 44.57±2.63 | 44.94±3.00 | 21.68±2.60 | 20.18±5.91 | 43.64±2.21 | 46.41±3.17 | 50.57±3.92 |
| | | 0.2 | 30.47±2.46 | 27.89±2.07 | 13.60±0.82 | 22.78±5.28 | 42.32±3.24 | 42.70±3.41 | 19.88±2.37 | 22.24±5.40 | 22.94±12.06 | 43.82±2.52 | 47.11±1.73 |
| | | 0.3 | 27.30±2.47 | 26.00±2.61 | 14.59±1.62 | 21.99±3.57 | 36.96±3.59 | 39.06±3.66 | 19.66±2.29 | 19.50±3.71 | 20.36±10.18 | 38.69±2.63 | 42.41±2.80 |
| | | 0.4 | 24.49±1.85 | 24.03±1.88 | 13.30±0.29 | 21.02±3.18 | 37.71±3.53 | 39.29±2.00 | 19.23±2.67 | 22.34±5.71 | 17.52±9.75 | 37.48±3.01 | 42.11±2.80 |
| | | 0.5 | 23.12±2.64 | 24.04±1.99 | 14.72±0.92 | 19.88±2.39 | 35.65±4.33 | 36.30±3.37 | 17.89±2.87 | 23.30±4.38 | 18.97±10.34 | 35.25±3.44 | 37.01±1.82 |
| | P12-L | - | 92.42±1.09 | 96.42±0.41 | 93.41±0.93 | 96.41±1.08 | 94.16±0.99 | 94.65±0.80 | 95.28±0.24 | 96.57±0.51 | 96.99±0.34 | 97.43±0.28 | 97.70±0.38 |
| | | 0.1 | 94.21±1.00 | 96.37±0.57 | 94.17±0.66 | 96.27±1.21 | 94.18±1.02 | 94.30±1.09 | 95.18±0.31 | 96.04±0.43 | 96.90±0.62 | 97.29±0.34 | 97.49±0.42 |
| | | 0.2 | 93.99±0.90 | 96.29±0.46 | 93.00±1.24 | 96.27±0.87 | 94.24±1.02 | 94.40±1.04 | 95.01±0.44 | 96.11±0.33 | 93.34±3.81 | 97.05±0.42 | 97.69±0.41 |
| | | 0.3 | 94.39±0.62 | 96.33±0.44 | 93.48±0.50 | 95.95±1.12 | 94.04±1.00 | 94.13±1.06 | 95.22±0.24 | 96.04±0.44 | 92.98±3.25 | 96.75±0.44 | 97.30±0.43 |
| | | 0.4 | 93.86±0.47 | 96.00±0.48 | 93.89±0.56 | 95.79±1.13 | 93.95±1.13 | 93.88±1.20 | 95.05±0.32 | 95.99±0.53 | 91.98±3.73 | 96.77±0.23 | 97.30±0.37 |
| | | 0.5 | 93.25±0.80 | 96.07±0.81 | 93.33±0.67 | 95.73±0.97 | 93.92±1.10 | 94.02±1.13 | 94.91±0.72 | 95.92±0.33 | 91.65±3.00 | 96.59±0.48 | 97.31±0.25 |
| | P19 | - | 31.24±4.15 | 31.12±5.25 | 47.37±2.97 | 41.13±8.01 | 29.60±6.26 | 28.05±6.23 | 30.34±1.80 | 50.63±3.32 | 41.12±3.30 | 57.96±4.10 | 41.16±3.02 |
| | | 0.1 | 25.59±1.80 | 8.05±9.56 | 41.98±3.41 | 33.40±6.12 | 28.90±5.93 | 25.12±5.46 | 28.37±2.22 | 45.75±4.62 | 34.69±4.55 | 54.45±4.23 | 40.59±1.60 |
| | | 0.2 | 25.44±1.50 | 7.32±8.16 | 37.80±2.37 | 25.89±7.49 | 30.25±6.21 | 25.53±5.53 | 24.70±2.79 | 43.42±3.78 | 7.80±7.43 | 51.56±3.96 | 39.28±1.67 |
| | | 0.3 | 26.26±3.09 | 6.30±5.47 | 33.01±1.41 | 19.71±5.26 | 31.07±6.91 | 26.57±6.17 | 22.15±1.58 | 43.79±4.57 | 7.40±6.01 | 50.24±3.16 | 36.86±1.51 |
| | | 0.4 | 22.38±4.02 | 5.95±4.66 | 29.33±5.31 | 14.87±3.98 | 31.99±5.50 | 27.18±5.31 | 18.30±1.67 | 44.01±4.23 | 5.46±2.79 | 49.05±4.06 | 35.27±1.33 |
| | | 0.5 | 23.65±2.54 | 5.35±3.61 | 23.58±4.49 | 11.84±2.87 | 32.90±5.68 | 28.28±6.02 | 17.34±0.85 | 43.15±4.06 | 6.94±3.24 | 46.12±3.99 | 32.42±1.47 |
| | PAM | - | 74.95±2.68 | 88.28±1.28 | 75.78±2.02 | 88.85±2.00 | 86.73±4.21 | 91.50±0.61 | 36.43±12.23 | 95.48±0.91 | 92.75±1.43 | 94.21±0.71 | 97.61±0.26 |
| | | 0.1 | 50.81±3.79 | 83.24±2.15 | 64.10±2.33 | 81.96±2.44 | 82.04±3.48 | 76.49±3.98 | 26.40±1.83 | 40.21±25.18 | 89.63±1.09 | 90.73±0.71 | 94.62±0.46 |
| | | 0.2 | 41.46±1.77 | 72.49±1.62 | 59.11±1.51 | 71.23±2.65 | 72.63±2.64 | 71.41±2.57 | 24.08±2.35 | 28.06±5.71 | 15.71±3.71 | 82.50±1.66 | 91.76±1.48 |
| | | 0.3 | 33.24±1.22 | 65.96±3.68 | 43.36±2.20 | 64.06±1.32 | 66.93±1.60 | 67.41±2.58 | 21.84±2.31 | 27.68±8.06 | 16.29±1.74 | 77.34±1.39 | 85.34±2.10 |
| | | 0.4 | 27.98±2.85 | 54.29±2.58 | 32.18±2.97 | 84.09±3.76 | 59.05±0.70 | 60.60±0.83 | 20.26±1.06 | 23.12±3.90 | 15.91±3.77 | 67.62±2.00 | 77.81±2.55 |
| | | 0.5 | 24.81±2.64 | 49.16±2.30 | 27.24±1.70 | 46.74±1.76 | 52.64±1.18 | 54.42±1.01 | 19.47±1.57 | 22.86±3.21 | 17.09±4.45 | 60.79±3.19 | 72.58±2.72 |

Table A4: Classification performance of ATENet and baselines when dropping variables with various ratios ∈ [0.1, 0.5] in the *leave-random-sensors-out* scenario. *Drop Ratio* denotes the ratio of missing variables.

| Metric | Dataset | w/o $\mathcal{L}_{TC}$ | w/o $\mathcal{L}_{TCP}$ | w/o $\mathcal{L}_{TCI}$ | w/o $\mathcal{L}_{VC}$ | ATENet |
|---|---|---|---|---|---|---|
| AUROC | P12-M | 85.53±1.29 | 85.61±1.17 | 85.62±1.20 | 85.49±1.17 | 85.54±1.26 |
| | P12-L | 79.86±1.95 | 79.81±2.22 | 79.56±1.96 | 79.08±2.92 | 79.64±2.41 |
| | P19 | 83.19±1.93 | 83.43±2.23 | 81.39±3.16 | 81.75±2.21 | 84.02±1.38 |
| | PAM | 99.02±0.23 | 99.01±0.22 | 99.08±0.30 | 98.84±0.52 | 99.18±0.15 |
| *Average Performance Drop Rate* (%) | | 0.20 | 0.13 | 0.91 | 3.22 | - |
| AUPRC | P12-M | 53.28±2.05 | 53.34±1.87 | 53.39±1.87 | 52.93±1.99 | 53.31±2.02 |
| | P12-L | 97.72±0.47 | 97.72±0.38 | 97.66±0.49 | 97.56±0.47 | 97.70±0.38 |
| | P19 | 36.75±3.75 | 38.87±5.24 | 35.08±5.48 | 29.32±4.18 | 41.16±3.02 |
| | PAM | 96.93±1.08 | 97.57±0.40 | 97.14±0.68 | 96.65±1.16 | 97.61±0.26 |
| *Average Performance Drop Rate* (%) | | 1.28 | 0.57 | 1.63 | 3.33 | - |

Table A5: Classification performance of ablations related to temporal and intervariable consistency regularization

| Method | P-12M | | P-12L | | P19 | | PAM | |
|---|---|---|---|---|---|---|---|---|
| | *AUROC* | *AUPRC* | *AUROC* | *AUPRC* | *AUROC* | *AUPRC* | *AUROC* | *AUPRC* |
| Fully Individualized Variant | 85.07 | 52.28 | 79.39 | **97.71** | 81.53 | 30.09 | 99.02 | 96.96 |
| ATENet (ours) | **85.54** | **53.31** | **79.64** | 97.70 | **84.02** | **41.16** | **99.18** | **97.61** |

Table A6: Classification performance of ATENet and fully individualized variant

### G.3.2 Comparison with Pretrain-then-Finetune Frameworks

Pretrain-then-finetune frameworks, such as ModernTCN [21], aim to learn general-purpose representations through task-agnostic pretraining followed by finetuning on downstream tasks. In contrast, ATENet adopts a task-specific learning strategy that directly optimizes representations for the classi-

| Method | P-12M | | P-12L | | P19 | | PAM | |
|---|---|---|---|---|---|---|---|---|
| | *AUROC* | *AUPRC* | *AUROC* | *AUPRC* | *AUROC* | *AUPRC* | *AUROC* | *AUPRC* |
| ModernTCN | 81.16 | 42.92 | 71.20 | 96.47 | 75.31 | 17.73 | 98.35 | 93.30 |
| ATENet | **85.54** | **53.31** | **79.64** | **97.70** | **84.02** | **41.16** | **99.18** | **97.61** |

Table A7: Classification performance of ATENet and ModernTCN

| Method | P-12M | | P-12L | | P19 | | PAM | |
|---|---|---|---|---|---|---|---|---|
| | *AUROC* | *AUPRC* | *AUROC* | *AUPRC* | *AUROC* | *AUPRC* | *AUROC* | *AUPRC* |
| FreRA | 73.63 | 33.39 | 68.28 | 96.14 | 78.36 | 31.60 | 93.94 | 76.31 |
| FGTI | 82.87 | 46.12 | 72.75 | 96.91 | 81.54 | **43.21** | **99.31** | 97.05 |
| ATENet | **85.54** | **53.31** | **79.64** | **97.70** | **84.02** | 41.16 | 99.18 | **97.61** |

Table A8: Classification performance of ATENet and frequency-domain approaches

fication objective. This approach can provide several advantages, including better task alignment, lower computational cost, and architectural simplicity.

To empirically evaluate the effectiveness of this design, we compared ATENet with ModernTCN, a recent pretrain-then-finetune method. As shown in the Table A7, ATENet consistently outperformed ModernTCN across all datasets in both AUROC and AUPRC, demonstrating that task-specific learning can be highly effective even without explicit pretraining.

### G.3.3 Comparison with Frequency-Domain Approaches

Frequency-domain approaches offer a complementary perspective on time-series analysis. However, their applicability to irregular multivariate time series is limited by the assumption of uniform sampling inherent in most Fourier-based methods [26, 39]. When this assumption is violated, as is often the case in irregular settings, frequency-based representations can become unstable or unreliable. This limitation likely contributes to the scarcity of related studies in this domain.

Nevertheless, we compared ATENet with two recent frequency-based methods: FreRA [37] and FGTI [43]. Although these methods were originally developed for regularly sampled data or different tasks, we adapted them to the irregular multivariate time-series classification setting to enable a fair comparison. As shown in the Table A8, ATENet outperformed both methods in most cases. While FGTI achieved comparable performance to ATENet in a few cases (e.g., AUPRC on P19), it relies on a diffusion-based architecture with significantly more parameters and longer processing time. In contrast, ATENet showed competitive results with a simpler and more efficient design.

## H  Robustness to Missing Observations

Learnable reference points and temporal consistency regularization can enhance the capability of capturing inherent temporal patterns by focusing on partial observations with inconsistent time intervals in irregular time series. That is, ATENet can robustly perform when some observations are missing along the time axis. To validate this effect, in Figure A1, we compared the AUROC of ATENet with that of MTSFormer and Raindrop, which showed the second-best and third-best performance in Table 1 of the main body, across various missing ratios $\in [0.1, 0.5]$. Here, we randomly hid observations in each sample of the test set along the time axis instead of using all observations for certain variables. As a result, ATENet consistently performed, unlike Raindrop, despite the presence of missingness along the time axis, demonstrating that our method effectively captures intricate temporal dependencies.

## I  Sensitivity Analysis

We analyzed hyperparameters affecting ATENet, including the time embedding function, the number of heads in MTA, the learning rate, and the loss weights. Here, we report classification performance using the AUPRC, which was used to select the best models for each dataset.

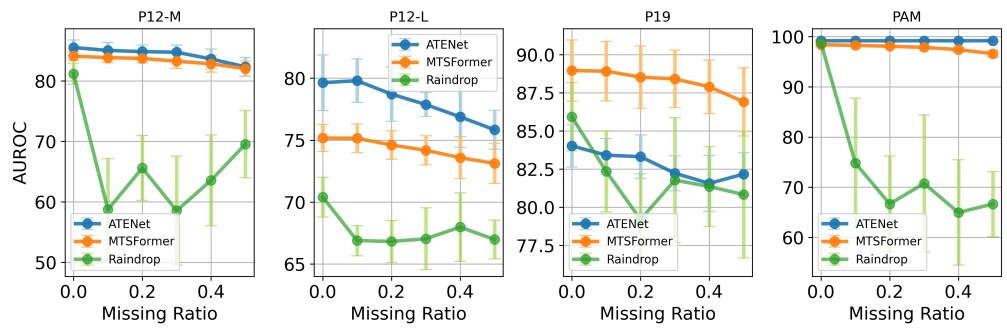

Figure A1: AUROC scores of ATENet and Raindrop across missing ratios ∈ [0.1, 0.5] along the time axis

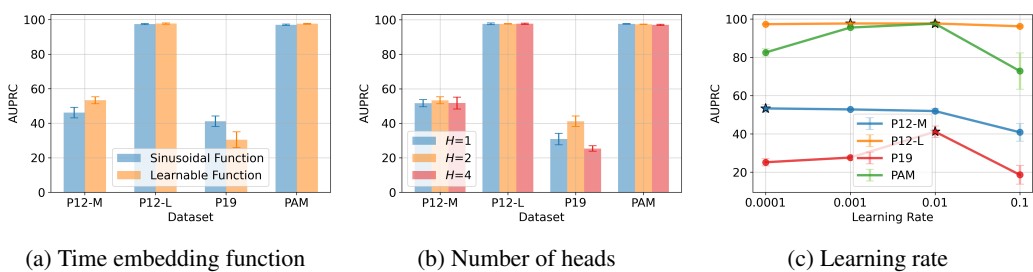

(a) Time embedding function      (b) Number of heads      (c) Learning rate

Figure A2: AUPRC scores according to (a) time embedding function, (b) number of heads, and (c) learning rate

**Time embedding function.** We analyzed the influence of the time embedding function $\phi$ by comparing the AUPRC for *sinusoidal* and *learnable* functions. As shown in Figure A2a, although the classification performance for the P12-L and PAM datasets was not sensitive to $\phi$, the classification performance on the P12-M and P19 datasets differed by about 5% depending on $\phi$. These performance gaps stem from how these embedding functions capture temporal dependencies and unique patterns in each dataset. The sinusoidal embedding function is fixed and periodic, thereby successfully capturing predictable and cyclic temporal patterns. Thus, it performs well in datasets with smooth or cyclic temporal structures. However, this function may struggle with non-periodic data, as it cannot adapt to complex time dependencies. In contrast, the learnable embedding function is more flexible in capturing non-periodic patterns and is able to learn temporal representations specialized to the data, making them advantageous for datasets with more complex or non-cyclic patterns. To sum up, the choice of time embedding function impacts classification performance based on the complexity of the temporal patterns in each dataset. Thus, exploring the optimal embedding function $\phi$ is necessary to enhance classification performance. In this study, we set $\phi$ to *sinusoidal* embedding function for the P19 dataset, whereas *learnable* one for the remaining datasets.

**Number of heads.** To examine the effect of the number of time embedding functions (attention heads), $H$, we compared the AUPRC on each dataset using various $H$s. As shown in Figure A2b, the classification performance slightly differed depending on $H$ in the P19 dataset. However, the performance differences for the other datasets were small, indicating that ATENet is not highly sensitive to the number of heads. We set the number of heads $H$ in the P12-M, P12-L, P19, and PAM datasets to 2, 4, 2, and 1, respectively.

**Learning rate.** In Figure A2c, we compared the AUPRC of ATENet by varying the learning rates. Consequently, the classification performance on the four datasets differed according to the learning rate. In this study, the learning rates for the P12-M, P12-L, P19, and PAM datasets were set to 0.0001, 0.001, 0.01, and 0.01, respectively.

**Loss weights.** We use two loss weights $\alpha$ and $\beta$ associated with temporal and intervariable consistency regularization terms to reflect intricate temporal and intervariable dependencies, respectively. Figure A3 exhibits the AUPRC of ATENet for various pairs of $\alpha$ and $\beta$. The performance differences for the P12-M, P12-L, and PAM datasets were relatively small, indicating ATENet's robustness

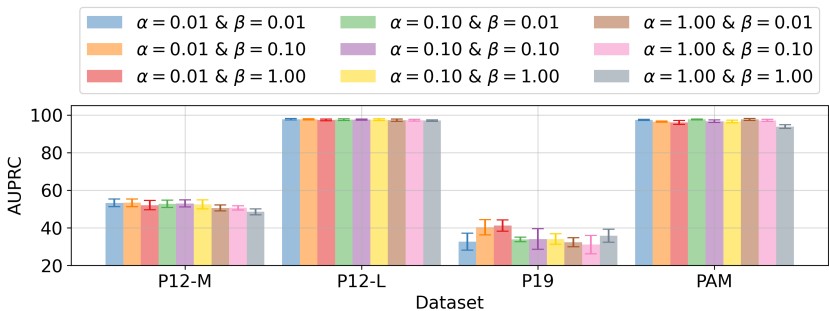

Figure A3: AUPRC scores according to weights for temporal and intervariable consistency regularization terms

against $\alpha$ and $\beta$. However, for the P19 dataset, the performance gap is significantly high. Therefore, in this study, we set a pair of $(\alpha, \beta)$ for the P12-M, P12-L, P19, and PAM datasets to (0.01, 0.1), (0.01, 0.01), (0.01, 1), and (0.1, 0.01) through a grid search for $\alpha \in \{0.01, 0.1, 1\}$ and $\beta \in \{0.01, 0.1, 1\}$.

## J   Code Availability

The code for reproducing our experimental results is available on GitHub at https://github.com/shlee-labs/ATENet.

