# OpenReview forum: "Adaptive Time Encoding for Irregular Multivariate Time-Series Classification"
_NeurIPS.cc/2025/Conference — NeurIPS 2025 poster_

### Official Review · Reviewer_ASQ7 · 2025-06-13

**Clarity:** 2
**Significance:** 2
**Originality:** 3
**Rating:** 5
**Confidence:** 3

**Summary:**

This paper proposes an adaptive time encoding method that generates latent representations at learnable reference points to improve irregular multivariate time-series classification. The approach enhances performance by capturing missingness patterns and incorporating temporal-intervariable relationships through consistency regularization.

**Questions:**

1. The Adaptive Time Encoding (ATE) proposed in the paper uses a set of learnable reference time points, but it does not provide details on how the hyperparameter K is determined. Does the choice of K significantly affect classification performance? If the target dataset has a different time distribution from the training set, do the learned reference time points still generalize well?
2. ATE uses a multi-head temporal attention mechanism for interpolation. However, in extremely sparse scenarios, this process may suffer from insufficient observation points, leading to uneven or even ineffective attention weight distributions. When the number of observations for an input variable is very limited, is the interpolation mechanism still effective?
3. Among the comparison methods, only one is from 2024. Are there any other more recent methods that could be included for comparison?
4. For Figure 6, could the raw data be provided as a reference? This would make the results more convincing.
5. The meanings of the horizontal and vertical axes in Figure 7 do not appear to be explained.

**Ethical Concerns:**

["NO or VERY MINOR ethics concerns only"]

**Final Justification:**

The author provided detailed answers to several of my questions, and I have no further questions. I don't think there are any other obvious issues with this paper and it can be accepted.

**Limitations:**

yes

**Paper Formatting Concerns:**

It would be clearer if the best results in the appendix tables were highlighted as they are in the main text.

**Quality:**

3

**Strengths And Weaknesses:**

Strengths: The paper is well-structured overall, with clear descriptions and rigorous analysis.
Weaknesses: Some parts lack sufficient detail or justification, which will be listed in the " Questions" section.

---

> ### Author Rebuttal · Authors · 2025-07-31
>
> We appreciate your helpful comments. We tried our best to address all of your comments, including five questions, **Q1-Q5**. We hope our responses help to clarify our work and answer all the concerns.
>
> ---
>
> ### **Q1. Choice of Number of Reference Points & Discussion on Effect of Distribution Shifts**
>
> - ***Choice of number of reference points***: To evaluate the impact of the number of reference points $K$, we conducted additional experiments with five different values of $K$: 16, 32, 64, 128, and 256. As shown in the table below, performance generally improves up to $K=128$, after which it plateaus or slightly drops. This suggests that a moderate number of reference points (e.g., 64-128) offers a good trade-off between representation capacity (expressiveness) and regularization. Notably, $K=128$ yields the best results on most datasets.
>
> | # Reference Points ($K$) |   P-12M   ||   P-12L   ||    P19     ||    PAM     ||
> |--------------------|---------|---------|---------|---------|----------|---------|----------|---------|
> |                    | *AUROC*   | *AUPRC*   | *AUROC*   | *AUPRC*   | *AUROC*    | *AUPRC*   | *AUROC*    | *AUPRC*   |
> | **16**             |  84.67    |  50.41    |  77.99    |  97.56    |   81.20    |  30.92    |   99.08    |  96.99    |
> | **32**             |  85.19    |  51.15    |  79.29    |  97.75    |   83.37    |  33.52    |   98.78    |  96.58    |
> | **64**             |  85.17    |  51.27    |  **79.65**    |  **97.81**    |   83.93    |  38.00    |   98.99    |  97.09    |
> | **128**            |  **85.54**    |  **53.31**    |  79.64    |  97.70    |   **84.02**    |  **41.16**    |  **99.18**    |  **97.61**    |
> | **256**            |  83.87    |  47.58    |  78.62    |  97.77    |   81.25    |  31.99    |   98.93    |  96.91    |
>
> - ***Discussion on effect of distribution shifts***: Regarding generalization to datasets with different temporal distributions (i.e., distribution shift), our experiments focused on within-dataset settings. However, we acknowledge that such shifts in temporal dynamics —such as differences in event density or measurement frequency between training and deployment environments—may affect the effectiveness of the learned reference points. In such cases, the learned reference points may no longer align with informative regions in the target data, potentially degrading interpolation quality and downstream performance. We consider this an important advanced direction and leave it for future work.
>
> ---
>
> ### **Q2. Extremely Sparse Scenarios**
>
> We agree that in extremely sparse scenarios, attention-based interpolation may become less effective due to insufficient observation points. However, this limitation is not unique to our approach; when observations are too sparse, no method is likely to perform well due to a fundamental lack of information.
>
> However, ATENet is designed to be more robust to such sparsity than conventional approaches by:
>
> - **Learning reference points** that adapt to regions with dense observations;
>
> - Using **attention-based interpolation** that flexibly aggregates information from even distant observations;
>
> - Introducing **temporal consistency regularization** based on random masking, which enhances representation quality even when data is partially missing.
>
> As shown in *Appendix H*, ATENet demonstrates strong robustness under high levels of missingness. Specifically, the model maintains competitive performance when up to 50% of the observations are randomly dropped, indicating that our method remains effective as long as a minimal level of temporal coverage is preserved.
>
> Nonetheless, we acknowledge that in extremely sparse settings, where few or no observations exist near any reference point, performance can degrade. We will include this limitation in Appendix F.
>
> ---
>
> ### **Q3. Additional Recent Baselines**
>
> To address your concern, we included ModernTCN [1], a recent and relevant baseline for irregular multivariate time-series classification, in our main experiments. As shown in the table below, ATENet consistently outperforms ModernTCN across all datasets in both AUROC and AUPRC, demonstrating the effectiveness of our approach.
>
> |  Dataset  | P-12M |           | P-12L |           |  P19  |           |  PAM  |           |
> |-------------|---------|---------|---------|---------|---------|---------|---------|---------|
> |             | *AUROC* | *AUPRC* | *AUROC* | *AUPRC* | *AUROC* | *AUPRC* | *AUROC* | *AUPRC* |
> | **ModernTCN** |   81.16   |   42.92   |   71.20   |   96.47   |   75.31   |   17.73   |   98.35   |   93.30   |
> |   **ATENet**  | **85.54** | **53.31** | **79.64** | **97.70** | **84.02** | **41.16** | **99.18** | **97.61** |
>
> ##### [1] Luo, D., & Wang, X. (2024, May). Moderntcn: A modern pure convolution structure for general time series analysis. In The twelfth international conference on learning representations (pp. 1-43).
>
> ---
>
> ### **Q4. Figure 6 – Raw Data Visualization**
>
> Thank you for the suggestion. We will revise *Figure 6* to include an overlay or side-by-side presentation of the corresponding raw time-series data alongside the attention weight visualizations. This update will allow for a more direct comparison between the observed values and the model’s learned attention focus.
>
> ---
>
> ### **Q5. Figure 7 – Axis Explanation**
>
> Thank you for pointing this out. In *Figure 7*, both the horizontal and vertical axes represent the variables in the multivariate time series. The matrices illustrate pairwise intervariable relationships captured from the input and learned representations. We will revise the figure caption to clearly indicate the meaning of the axes and improve clarity for the reader.

---

> > ### Comment · Reviewer_ASQ7 · 2025-08-05
> >
> > Thank you very much for the author's detailed response! I have carefully read the reply and have no further questions. I will change the score from 4 to 5.

---

> > > ### Author Response · Authors · 2025-08-05
> > >
> > > Thank you very much for your thoughtful reconsideration and for updating the score. We sincerely appreciate your time and constructive feedback.

---

### Official Review · Reviewer_x3TD · 2025-07-01

**Clarity:** 3
**Significance:** 2
**Originality:** 2
**Rating:** 4
**Confidence:** 3

**Summary:**

The authors address the problem of classifying irregular time
series with missing values (IMTS). They propose a model
that for each channel interpolates the observations to a
reference time points, that are initialized in regular intervals,
but then can be updated. Cross attention is used for
the interpolation. Finally latent representations are
added across attention heads and channels and the class
probabilities regressed on this. The loss is combined with
- i) a contrastive loss for the reference point representations
  of original instances and masked instances vs. other
  instances,
- ii) a contrastive loss for the reference point representations
  of original instances and masked instances vs. other
  reference time points (of the same instance), and
- iii) a cross entropy loss between channels.

In experiments on four datasets against many baselines
they show that their model outperforms existing
methods.

**Questions:**

- q1. how did you optimize your hyperparameters? (w1)

**Ethical Concerns:**

["NO or VERY MINOR ethics concerns only"]

**Final Justification:**

--- added after rebuttal
- the authors' reply that their hyperparameters are optimited
  on the validation sample resolves my main concern.
- as overall lifts are small, I will update my score just by 1.

**Limitations:**

yes

**Quality:**

3

**Strengths And Weaknesses:**

strong points:
- s1. plausible and simple model: interpolate channels to an adaptive
  list of reference points.
- s2. good, mostly consistent lifts over the baselines.

weak points:
- w1. hyperparameters maybe trained on test?

ad w1. hyperparameters maybe trained on test?
- how did you optimize your hyperparameters?
  in appendix I you say " Here, we report classification
  performance using the AUPRC, which was used to
  select the best models for each dataset." (line 635).
  Does this mean you optimized your hyperparameters
  on test? If so, would your results still stand if you had
  to pick them on a validation sample within train?
  Are the hyperparameters of the baseline models
  also optimized on test?

---

> ### Author Rebuttal · Authors · 2025-07-30
>
> Thank you for your comments. We have carefully addressed your concern, including the weakness (**W1**) and its corresponding question (**Q1**). We hope our response helps clarify our work and answer your concerns.
>
> ---
>
> ### **W1 & Q1. Optimization on Hyperparameters**
>
> We clarify that no hyperparameters were tuned on the test set. As described in *Appendix C*, each dataset was split into training (80%), validation (10%), and test (10%) sets. All hyperparameters were selected based on performance on the validation set, and the final test results were obtained using the model that achieved the best validation performance.
>
>  We acknowledge that the explanation in *Appendix I* may have caused confusion, and we will revise the wording to avoid any potential misunderstanding. Note that this procedure was applied consistently to all baseline models as well.

---

> ### Author Response · Authors · 2025-08-06
>
> As the discussion deadline is approaching, we just wanted to check in and see if you had any further questions or comments on our paper. We’d be happy to address them!

---

### Official Review · Reviewer_7wnM · 2025-07-02

**Clarity:** 3
**Significance:** 2
**Originality:** 3
**Rating:** 4
**Confidence:** 4

**Summary:**

This paper proposes ATENet, an end-to-end encoder-classifier framework for irregular multivariate time-series classification. The key contribution is a novel Adaptive Time Encoding (ATE) module that learns reference time points to generate latent representations while capturing irregular sampling patterns and missingness. To further enhance the learned representations, the framework introduces two regularization terms: temporal consistency (capturing fine-grained temporal dynamics) and intervariable consistency (modeling dependencies across variables efficiently without graph neural networks). The proposed method is evaluated across multiple real-world benchmarks and demonstrates strong performance and computational efficiency compared to state-of-the-art methods.

**Questions:**

Pretraining vs. End-to-End: Have you considered comparing ATENet with pretrain-then-finetune frameworks (e.g., PrimeNet)? Your method does representation learning implicitly; how does it perform against explicit representation learning approaches?

**Ethical Concerns:**

["NO or VERY MINOR ethics concerns only"]

**Limitations:**

The storyline, challenges and corresponding technical component, is complete, but to somehow incremental and not exciting.

**Quality:**

2

**Strengths And Weaknesses:**

Strengths:
1.The paper identifies two concrete challenges in modeling irregular multivariate time series—(1) failure to utilize missingness patterns, and (2) neglect of inter-variable correlations—and addresses each with dedicated technical components.
2.The method is tested on diverse datasets (healthcare and activity monitoring), and the experiments span multiple evaluation setups including robustness to missing variables and computational efficiency.
3.The paper is easy to follow, with clear visualizations and ablation studies that support the claims.

Weakness:
1.  While the paper includes ablation studies, it remains unclear which component (adaptive reference points, temporal consistency, or intervariable consistency) contributes the most in specific cases.
2. The paper combines known ideas (learnable queries, contrastive consistency, and interpolation attention) in a cohesive way, but may feel more incremental rather than fundamentally new to readers familiar with this area.

---

> ### Author Rebuttal · Authors · 2025-07-31
>
> We appreciate your helpful comments. We tried our best to address all of your comments, including two weaknesses, **W1** and **W2**, and one question, **Q1**. We hope our responses help to clarify our work and answer all the concerns.
>
> ---
>
> ### **W1. Ablation Studies**
>
> As shown in *Table 2* of the manuscript, the use of adaptive reference time points consistently improves classification performance across all datasets compared to fixed reference points (i.e., regular, sparse, or dense), highlighting their effectiveness in capturing irregular sampling patterns.
>
> Additionally, *Table 3* and *Appendix G.2* demonstrate that both temporal and intervariable consistency contribute meaningfully to overall performance, with their relative impact varying depending on dataset characteristics. Specifically, removing temporal consistency results in an average AUPRC drop of 1.28%, whereas removing intervariable consistency leads to a larger average drop of 3.33%. These findings indicate that both components are beneficial, with intervariable consistency showing a more substantial effect in our experimental settings.
>
> We will revise Section 4.3 to more clearly highlight the component-wise contributions.
>
> ---
>
> ### **W2. Novelty of Our Work**
>
> While ATENet builds upon familiar components such as attention-based interpolation and contrastive regularization, its key novelty lies in how these components are effectively integrated and optimized to address the unique challenges of irregular multivariate time-series classification.
>
> - **Adaptive encoding process**: We propose a novel encoding framework that learns reference time points in an end-to-end supervised manner, eliminating the need for predefined anchors or handcrafted temporal discretization. Unlike previous approaches, our method directly optimizes the reference points with respect to both the task objective and the distribution of observed timestamps. This design enables the model to align irregular sequences in a task-aware and data-adaptive temporal space, thereby improving both the expressiveness and generalization ability of the learned representations across varying sampling patterns.
>
> - **Consistency regularization**: To further improve the quality and robustness of representations, ATENet incorporates two lightweight yet effective regularization techniques:
>
>   - *Temporal consistency regularization*: This component enforces stability in the learned representations under random temporal masking—a perturbation strategy tailored for sparse or irregular time series. By promoting invariance to partially missing observations, it enhances generalization under temporal noise or missingness.
>
>   - *Intervariable consistency regularization*: We propose a novel contrastive objective that encourages structural consistency across variables by exploiting the outer product between input and representation spaces. This approach efficiently captures intervariable dependencies and serves as a lightweight alternative to more complex graph-based or recurrent models, enhancing cross-variable coherence without incurring high computational overhead.
>
> We will revise the introduction to more clearly emphasize the key novelties of our approach.
>
> ---
>
> ### **Q1. Comparison with Pretrain-then-Finetune Frameworks**
>
> Pretrain-then-finetune frameworks, such as PrimeNet [1] and ModernTCN [2], aim to learn general-purpose representations through task-agnostic pretraining followed by finetuning on downstream tasks. In contrast, ATENet adopts a task-specific learning strategy that directly optimizes representations for the classification objective. This approach can provide several advantages, including better task alignment, lower computational cost, and architectural simplicity.
>
> To empirically evaluate the effectiveness of this design, we compared ATENet with *ModernTCN*, a recent pretrain-then-finetune method. As shown in the table below, ATENet consistently outperformed ModernTCN across all datasets in both AUROC and AUPRC, demonstrating that task-specific learning can be highly effective even without explicit pretraining.
>
> |  Dataset  | P-12M |           | P-12L |           |  P19  |           |  PAM  |           |
> |-------------|---------|---------|---------|---------|---------|---------|---------|---------|
> |             | *AUROC* | *AUPRC* | *AUROC* | *AUPRC* | *AUROC* | *AUPRC* | *AUROC* | *AUPRC* |
> | **ModernTCN** |   81.16   |   42.92   |   71.20   |   96.47   |   75.31   |   17.73   |   98.35   |   93.30   |
> |   **ATENet**  | **85.54** | **53.31** | **79.64** | **97.70** | **84.02** | **41.16** | **99.18** | **97.61** |
>
> In the original manuscript, we did not include pretrain-then-finetune frameworks because our study focuses on fully supervised learning using only task-specific data, and most baseline methods also follow this setting, ensuring fair and consistent comparisons.
>
> We acknowledge the importance of general-purpose representation learning and consider bridging task-specific and task-agnostic strategies a promising direction for future work. We will also include the additional results and discussion in the revised manuscript.
>
> ##### [1] Chowdhury, R. R., Li, J., Zhang, X., Hong, D., Gupta, R. K., & Shang, J. (2023, June). Primenet: Pre-training for irregular multivariate time series. In Proceedings of the AAAI Conference on Artificial Intelligence (Vol. 37, No. 6, pp. 7184-7192).
>
> ##### [2] Luo, D., & Wang, X. (2024, May). Moderntcn: A modern pure convolution structure for general time series analysis. In The twelfth international conference on learning representations (pp. 1-43).

---

> > ### Comment · Reviewer_7wnM · 2025-08-05
> >
> > Thanks for further clarification about the framework novelty and providing extra experiments. I hope in the revised version, authors could compare with irregular-specific methods rather than general methods which typically assume data is continuous and regularly sampled.
> >
> > I will remain my score and hope these clarification will be added into the revised manuscript.

---

> > > ### Author Response · Authors · 2025-08-06
> > >
> > > Thank you for the thoughtful comments.
> > >
> > > Our experimental setup included a range of representative methods that explicitly handle irregularly sampled time series, such as mTAND, SeFT, Raindrop, Warpformer, and MTSformer. We also considered methods that focus on missing-value handling in time series, such as DGM² and GRU-D, which model informative missingness patterns. In addition, we evaluated against general-purpose sequence models (e.g., Transformer, ModernTCN) to provide context on how ATENet compares with strong baselines that do not explicitly address irregular sampling.
> > >
> > > Across these comparisons, ATENet consistently outperformed the baselines, demonstrating both its competitiveness in irregular-specific settings and its robustness relative to widely used general models.
> > >
> > > In the revised manuscript, we will explicitly categorize the baselines into irregular-specific and general-purpose groups in Section 4.1 to make this clearer. We appreciate your constructive feedback and will ensure these clarifications are clearly reflected in the updated version.

---

### Official Review · Reviewer_8fHo · 2025-07-03

**Clarity:** 3
**Significance:** 2
**Originality:** 3
**Rating:** 3
**Confidence:** 4

**Summary:**

This paper investigates the challenging temporal classification task with irregularly measured time-series as inputs. The authors propose a novel interpolation-based encoder-classifier structure, ATENet, with learnable reference points for the time-series embedding. In addition to the gradient from classification loss, two additional regularization terms are introduced in ATENet to enhance the representation quality of embeddings from the encoder. Specifically, with randomly masked data points, the instance-level and point-level contrastive losses are utilized to encourage the encoder to effectively capture unique temporal characteristics of the input time-series. In the meantime, the consistency loss enables ATENet to preserve temporal dependency structure of the input in its latent embeddings in the sense of outer products. The high-quality time-series representation learned by the encoder of ATENet enables a straight-forward design of the classifier network which greatly contributes to the improved computation efficiency compared to transformer- or graph-based approaches. Experimental results on three real-world datasets show that ATENet achieve better time-series classification accuracy over multiple state-of-the-art approaches. The authors also provide detailed ablation study to demonstrate the effectiveness of their different design choices in ATENet.

**Questions:**

1. What is the implicit assumption on time-series sparsity in AETNet? Is AETNet applicable to any time-series data?
2. Are the missing patterns always important to time-series classification? What is the performance of AETNet on datasets with missingness at random?
3. Does the time-series interpolation in Eq. 4 introduce inductive biases to AETNet? Would it be affected by noisy signals?
4. Is the set of key reference points $\mathbf{r}$ globally shared by all samples in a dataset? What is the impact of this design?
5. Is the outer product in Eq. 10 sufficient to model complex temporal patterns and dependencies across time steps, especially when the time-series are irregularly measured?
6. There should be additional baselines with time-series embeddings in the frequency domain.
7. ATENet consistently has lower performance than MTSFormer on P19 datasets according to Table 1 and Figure 4. There should be analysis on the underlying reasons and potential limitations of ATENet.

**Ethical Concerns:**

["NO or VERY MINOR ethics concerns only"]

**Final Justification:**

My major concerns with the global reference points of ATENet and the associated challenges from time-series alignment remain unresolved after the rebuttal. In contrary to many modern sequential models like RNNs, Transformers and Mamba, ATENet's time-series interpolation mechanism is strongly coupled with the learned global reference points and may fail to generalize on novel data points due to distributional shifts. Therefore, I tend to maintain my rating for this work.

**Limitations:**

yes

**Paper Formatting Concerns:**

No obvious issue found.

**Quality:**

3

**Strengths And Weaknesses:**

Strengths
- ATENet enables efficient processing of irregularly measured time-series through interpolation at key reference points and fixed length representation of predictive temporal patterns. Further, the reference points could be flexibly learned based on the training data instead of being manually designed which usually requires domain knowledge and expertise.
- The usage of positional encoding of observation time $t$ and reference point $r$ as key and query in attention mechanism is interesting and might be potentially equivalent to the concept of dynamic time warping (DTW) which is well-known to be effective for comparison of time-series of variable lengths.
- The notion of temporal consistency between the latent embedding $z$ and input time-series $x$ is sort of novel, and the design of corresponding regularization terms indeed have huge contribution to the classification performance according to the ablation study results in Table 3. The high-quality latent representation also enables a simplified classifier model, which contributes to the improvement in computational efficiency of ATENet compared to other models.

Weaknesses
- The fundamental assumption on the sparsity of the time-series data is not explicitly discussed. For instance, for meaningful interpolation of the irregularly measured time-series in Eq. 4, the sampling frequency of the input signals should satisfy certain criteria. If the measurement is overly sparse, ATENet may fail to distinguish different input time-series due to the lack of effective data points.
- The authors should also discuss any potential inductive biases introduced by the attention-based interpolation described in Section 3.2.
- It is unclear if the set of reference points $\mathbf{r}$ is globally shared by all samples or dynamically computed for individual input time-series. The authors need to justify for a global set of reference point $\mathbf{r}$ if applicable.
- Relevant study applying Fourier transform and spectral analysis should be discussed in literature review, and there should be additional baselines (utilizing frequency domain representations) from these researches.
- The performance gain of AETNet seems to be marginal except for AUPRC on P12-M and PAM dataset.

---

> ### Author Rebuttal · Authors · 2025-07-31
>
> Thank you for your thoughtful comments. We tried our best to address all of your comments, including five weaknesses, **W1-W5**, and six questions, **Q1-Q6**. We hope our responses help to clarify our work and answer all the concerns.
>
> ---
>
> ### **W1 & Q1. Assumptions on Time-Series Sparsity**
>
> ATENet implicitly assumes that the input time-series data are not excessively sparse. When observations are too infrequent or occur far from any reference points, the interpolation process becomes unreliable. This can lead to latent representations that fail to capture meaningful temporal patterns, thereby reducing their discriminative ability for downstream tasks.
>
> Importantly, ATENet does not assume uniform sampling of multivariate time series. In many real-world scenarios, such as clinical or industrial time series, observations are recorded irregularly, often reflecting meaningful event dynamics (e.g., dense measurements near symptoms or anomalies). ATENet is explicitly designed to leverage such irregularities as informative signals by:
>
> - **Learning reference points** that adapt to data-driven observation patterns, especially around informative regions;
>
> - Using **attention-based interpolation** that effectively aggregates information from both nearby and even distant observations;
>
> - Introducing **temporal consistency regularization** based on random masking, which enhances representation quality even when data is partially missing.
>
> This design enables ATENet to handle irregular and sparse inputs without relying on strict assumptions (e.g., Nyquist sampling).
>
> To empirically assess this robustness, we conducted experiments in Appendix H by randomly dropping observations along the time axis. The results show that ATENet maintains competitive performance even under relatively high missing ratios (up to 50%), demonstrating its effectiveness under considerable levels of sparsity.
>
> ---
>
> ### **Q2. Are Missing Patterns Always Important?**
>
> While missingness patterns are not always informative, they often reflect underlying processes in real-world applications, particularly in healthcare settings, where critical conditions often trigger more frequent measurements. The proposed method is designed to leverage such patterns when they are informative, but does not rely on them exclusively.
>
> We will revise the manuscript to clarify this point more explicitly.
>
> ---
>
> ### **W2 & Q3. Inductive Biases and Robustness to Noise**
>
> Our attention-based interpolation is built upon a set of learnable reference time points and a scaled dot-product attention mechanism over time embeddings. This design inherently induces a form of temporal locality bias: observations that are temporally closer to a reference point receive higher attention weights, similar to kernel smoothing [1-3].
>
> However, unlike fixed kernels or handcrafted interpolation schemes, our method learns both the reference points and time embeddings in an end-to-end manner. This adaptive design allows the model to flexibly capture irregular temporal patterns in the data, reducing reliance on rigid inductive assumptions while still promoting attention to informative temporal regions—such as those surrounding clinical events or machine anomalies—where observations tend to be dense.
>
> This inductive bias can have nuanced effects in the presence of noise. On one hand, it may help suppress temporally isolated noise by assigning it low attention weights. On the other hand, noise that occurs near densely sampled regions may still receive relatively high attention, potentially degrading interpolation quality.
>
> Although ATENet incorporates temporal and intervariable consistency regularization to mitigate such issues, we acknowledge that extreme noise remains a challenge (as discussed in *Appendix F*). We therefore consider integrating noise-aware extensions, such as Kalman filtering or denoising transforms, as a promising direction for future work.
>
> ##### [1] Li, S., Jin, X., Xuan, Y., Zhou, X., Chen, W., Wang, Y. X., & Yan, X. (2019). Enhancing the locality and breaking the memory bottleneck of transformer on time series forecasting. Advances in neural information processing systems, 32.
>
> ##### [2] Niu, P., Zhou, T., Wang, X., Sun, L., & Jin, R. (2024). Attention as robust representation for time series forecasting. arXiv preprint arXiv:2402.05370.
>
> ##### [3] Choromanski, K., Likhosherstov, V., Dohan, D., Song, X., Gane, A., Sarlos, T., ... & Weller, A. (2020). Rethinking attention with performers. arXiv preprint arXiv:2009.14794.
>
> ---
>
> ### **W3 & Q4. Globally Shared Reference Points**
>
> The set of reference time points is shared globally across all samples within a dataset and is learned as task-adaptive parameters during training. Rather than being fixed or predefined, these reference points are optimized to align with common yet task-relevant temporal patterns across the dataset. This enables the model to perform meaningful interpolation even in the presence of irregular or sparse observations.
>
> Although the reference points are shared, the attention-based interpolation weights are computed individually for each sample, based on the unique observation timestamps of that sequence. This design allows ATENet to retain sample-specific adaptability while leveraging a global temporal structure that facilitates efficient computation and stable optimization, particularly beneficial under sparse or noisy conditions.
>
> We will clarify this point explicitly in Section 3.2.1 of the revised manuscript.
>
> ---
>
> ### **W4 & Q5. Comparison with Frequency-Domain Approaches**
>
> Frequency-domain approaches offer a complementary perspective on time-series analysis. However, their applicability to irregular multivariate time series is limited by the assumption of uniform sampling inherent in most Fourier-based methods [1, 2]. When this assumption is violated—as is often the case in irregular settings—frequency-based representations can become unstable or unreliable. This limitation likely contributes to the scarcity of related studies in this domain.
>
> Nevertheless, to address your concern, we compared ATENet with two recent frequency-based methods: *FreRA* [3] and *FGTI* [4]. Although these methods were originally developed for regularly sampled data or different tasks, we adapted them to the irregular multivariate time-series classification setting to enable a fair comparison.
>
> As shown in the table below, ATENet outperforms both methods in most cases. While *FGTI* achieves comparable performance to ATENet in a few cases (e.g., AUPRC on P19), it relies on a diffusion-based architecture with significantly more parameters and longer processing time. In contrast, ATENet achieves competitive results with a simpler and more efficient design.
>
> | Dataset | P-12M |           | P-12L |           |  P19  |           |  PAM  |           |
> |-----------|---------|---------|---------|---------|---------|---------|---------|---------|
> |  | *AUROC* | *AUPRC* | *AUROC* | *AUPRC* | *AUROC* | *AUPRC* | *AUROC* | *AUPRC* |
> |  **FreRA**  |   73.63   |   33.39   |   68.28   |   96.14   |   78.36   |   31.60   |   93.94   |   76.31   |
> |   **FGTI**  |   82.87   |   46.12   |   72.75   |   96.91   |   81.54   | **43.21** | **99.31** |   97.05   |
> |  **ATENet** | **85.54** | **53.31** | **79.64** | **97.70** | **84.02** |   41.16   |   99.18   | **97.61** |
>
> We will expand the literature review to include these frequency-based approaches and explicitly discuss their limitations under irregular sampling.
>
> ##### [1] Reschke, M., Kunz, T., & Laepple, T. (2019). Comparing methods for analysing time scale dependent correlations in irregularly sampled time series data. Computers & Geosciences, 123, 65-72.
>
> ##### [2] Wang, H., Pan, L., Shen, Y., Chen, Z., Yang, D., Yang, Y., ... & Tao, D. FreDF: Learning to Forecast in the Frequency Domain. In The Thirteenth International Conference on Learning Representations.
>
> ##### [3] Tian, T., Miao, C., & Qian, H. (2025). FreRA: A Frequency-Refined Augmentation for Contrastive Learning on Time Series Classification. arXiv preprint arXiv:2505.23181.
>
> ##### [4] Yang, X., Sun, Y., & Chen, X. (2024). Frequency-aware generative models for multivariate time series imputation. Advances in Neural Information Processing Systems, 37, 52595-52623.
>
>
>
> ---
>
> ### **W5 & Q6. Marginal Performance Gain**
>
> We acknowledge that the performance gains of ATENet are relatively marginal on some datasets, especially the P19 dataset. Nonetheless, ATENet achieves superior classification performance on other datasets while offering substantial improvements in computational efficiency, requiring significantly fewer parameters and achieving faster training times (see *Figure 5*). This trade-off between performance and efficiency can be particularly valuable in resource-constrained settings.
>
> As discussed in *Appendix F*, the P19 dataset may exhibit extreme variability and noise in sampling times and values, making it difficult for interpolation-based methods, such as mTAND and ATENet, to capture subtle temporal patterns effectively and generalize the reference time points for all data.
>
> We will revise Section 4.2 and Appendix F to better highlight this trade-off and explicitly acknowledge such limitations in challenging settings.
>
> ---
>
> ### **Q4. Role of Outer Product**
>
> The outer product in Eq. (10) is not designed to capture temporal dependencies, but rather to model intervariable relationships within multivariate time series. Specifically, it encourages consistency between the structural relationships observed in the input variables and those encoded in the learned representations. This mechanism helps the model preserve meaningful cross-variable dependencies, which are particularly important in multivariate and high-dimensional settings.

---

> > ### Comment · Reviewer_8fHo · 2025-08-03
> >
> > I appreciate the detailed responses by the authors and the new experimental results. I still have concerns regarding the global reference points.
> >
> > ### **Globally Shared Reference Points**
> > While the reference time points of ATENet are learned dynamically on each dataset, in different event sequences, the set of optimal reference points may differ, leading to suboptimal accuracy of ATENet. Also, this may pose the time-series alignment a significant issue. For instance, two similar time-series may have similar sequences of temporal events but with measurements starting at different periods. The global reference points may prevent ATENet from correctly predicting future events for the two sequences.

---

> ### Author Response · Authors · 2025-08-05
> **Re: Globally Shared Reference Points**
>
> Thank you for the insightful follow-up.
>
> We acknowledge the concern that globally shared reference time points may lead to suboptimal interpolation when individual sequences exhibit temporal shifts, even if they share similar underlying patterns. However, several key aspects of ATENet mitigate this issue:
>
> **1. Learned reference structure with sample-specific interpolation**:
>
> ATENet uses a globally shared set of reference time points, which are not fixed but jointly learned with model parameters to capture task-relevant temporal structures across the training data. These reference points act as soft anchors that reflect typical patterns, such as common event timings, even in the presence of irregular or misaligned sequences. That is, when the training data contains diverse alignment distributions, the learned reference points are flexibly positioned to reflect such variations.
>
> Importantly, although the reference points are shared, the attention-based interpolation is computed individually for each sample based on its actual observation timestamps. This design enables ATENet to adaptively align each sequence with the learned temporal structure, effectively handling variability in starting times, observation frequencies, or event timing.
>
> As long as the number of reference points is set to be sufficiently expressive, this shared-yet-adaptive mechanism allows the model to handle a wide range of alignment scenarios without requiring per-sample customization of the reference set.
>
> **2. Empirical tradeoff - stability vs. flexibility**:
>
> Following the comment, we evaluated a variant that computes reference points separately for each sample. Although this approach offers better flexibility, it can be more sensitive to noise and irregular sampling, often resulting in unstable training and degraded generalization performance.
>
> As shown in the table below, the global shared design (ours) showed comparable or slightly better performance across most datasets. Notably, the sample-specific variant showed a substantial drop in performance on the P19 dataset, which may contain highly variable and noisy sequences (see Appendix F). This highlights that globally shared reference points, combined with sample-specific attention-based interpolation, offer better robustness under challenging conditions.
>
> | **Method**                 | **P-12M** |      | **P-12L** |      | **P19**   |      | **PAM**   |      |
> |------------------------|-------|------|--------|------|--------|------|--------|------|
> |                        | *AUROC* | *AUPRC* | *AUROC* | *AUPRC* | *AUROC* | *AUPRC* | *AUROC* | *AUPRC* |
> | **Sample-specific variant**| 85.07 | 52.28 | 79.39 | **97.71** | 81.53 | 30.09 | 99.02 | 96.96 |
> | **ATENet (ours)**                   | **85.54** | **53.31** | **79.64** | 97.70 | **84.02** | **41.16** | **99.18** | **97.61** |
>
>
> **3. Scope and distribution shift**:
>
> If an input sequence exhibits temporal structures not seen in the training data, such as entirely novel patterns, this constitutes a form of distribution shift problem. While this is an important direction for future research, it is beyond the scope of our current work.
>
> That said, ATENet generalizes well as long as the training data adequately reflects the temporal variations likely to be encountered during inference.
>
>
> In the revised manuscript, we will clarify this design choice in Section 3.2.1 and elaborate on the empirical findings and limitations in Appendix F.

---

> > ### Comment · Reviewer_8fHo · 2025-08-05
> >
> > Thanks for the response.
> >
> > > As long as the number of reference points is set to be sufficiently expressive, this shared-yet-adaptive mechanism allows the model to handle a wide range of alignment scenarios without requiring per-sample customization of the reference set.
> >
> > Given that the events/stages/patterns in time-series samples may not be readily aligned, if we reasonably assume that the key time points in a dataset is uniformly distributed, then maybe the set of optimal reference points is simply an evenly spaced points.

---

> > > ### Author Response · Authors · 2025-08-05
> > >
> > > We appreciate your thoughtful follow-up.
> > >
> > > We agree that if the key events or stages in a time series are indeed uniformly distributed, then a set of evenly spaced reference points could serve as a reasonable baseline.
> > >
> > > However, in many real-world scenarios, event timings are often non-uniform or governed by mixtures of distributions, rather than being uniformly distributed. In such cases, uniformly spaced reference points may be suboptimal, as they can fail to adequately capture regions where important patterns are concentrated. In contrast, learnable reference points can adapt to these informative regions, especially when key temporal patterns vary in frequency or density.
> > >
> > > Our approach learns these reference points in a data-driven manner, allowing them to capture the underlying temporal structures and missingness patterns without manual tuning. This flexibility can enhance representation quality, even when time points across samples are misaligned.
> > >
> > > As shown in *Table 2* of the main text, the learned reference points consistently outperform ablation models with fixed spacing (*Regular*, *Sparse*, *Dense*), confirming that learning the reference set is beneficial for modeling irregular and misaligned sequences.

---

> > > > ### Comment · Reviewer_8fHo · 2025-08-05
> > > >
> > > > Thanks for the further clarification. I've checked Table 2 in the main text again. While there are certain degrees of improvement of ATENet compared to ablations with regular/sparse/dense reference time points, the performance gain is sort of marginal. I would suggest the authors to add appropriate justification for using learnable global reference points in future versions of their paper.
> > > >
> > > > My ultimate concern with the global reference points is about time-series alignment. Even if the distribution of event timings indeed show certain pattern in the dataset, it may be highly related to the data curation process. In practical scenarios, the measurement of a signal can start at any stage, which results in distributional shift in the optimal reference points. The authors are welcome to provide justification for this if there is evidence that the starting points of time-series samples are fully predictable.

---

> > > > > ### Author Response · Authors · 2025-08-06
> > > > >
> > > > > Thank you for the additional comments.
> > > > >
> > > > > **How alignment is handled in ATENet**
> > > > >
> > > > > To address the concern about arbitrary start times, we first clarify how alignment is performed in ATENet. Although reference points are globally shared, alignment is performed per sequence via *timestamp-aware attention-based interpolation*. The attention mechanism operates within each realized observation window, identifying the positions of key events relative to the learned reference points. This enables the model to focus on the most informative regions even when sequences start at different times [1, 2].
> > > > >
> > > > > **On start-time predictability**
> > > > >
> > > > > We note that ATENet does not assume that start times are fully predictable. However, in many real-world domains, such as clinical records, manufacturing logs, or sensor monitoring, measurements are often tied, implicitly or explicitly, to trigger events (e.g., symptom onset, process start, scheduled inspections). Even if starting points are not fully predictable, latent patterns may exist, and learning global reference points allows the model to capture and exploit such patterns for better interpolation and downstream prediction [3, 4].
> > > > >
> > > > > **Global vs. local “uniformity”**
> > > > >
> > > > > Apparent uniformity of key-event timing often emerges only when aggregating all samples into a single global timeline. At the instance or window level, key events are frequently non-uniform, with certain regions being more informative [5, 6]. Our learned reference points, in conjunction with the attention mechanism, adapt to capture these informative regions when they exist. Conversely, when no positional preference is present, the model naturally assigns reference points and attention weights in a near-uniform manner, ensuring that performance does not degrade.
> > > > >
> > > > > **Why use globally learned reference points**
> > > > >
> > > > > To summarize, this design choice is motivated by three key benefits:
> > > > >
> > > > > - Robustness and stability: Mitigates overfitting to per-sample noise and outliers, unlike fully sample-specific reference points.
> > > > >
> > > > > - Generalization: Encodes dataset-level temporal structure while allowing per-sample adaptability through attention.
> > > > >
> > > > > - Efficiency: Reduces computational cost compared to per-sample optimization without losing flexibility.
> > > > >
> > > > > **Scope and future work**
> > > > >
> > > > > We acknowledge that in deployments where start-time distributions differ drastically from training data, this constitutes a distribution shift problem [7, 8]. While addressing such shifts is beyond the scope of our work, future work could extend ATENet to adapt reference points to such problems.
> > > > >
> > > > >
> > > > > ##### [1] Li, S., Jin, X., Xuan, Y., Zhou, X., Chen, W., Wang, Y. X., & Yan, X. (2019). Enhancing the locality and breaking the memory bottleneck of transformer on time series forecasting. Advances in neural information processing systems, 32.
> > > > > ##### [2] Niu, P., Zhou, T., Wang, X., Sun, L., & Jin, R. (2024). Attention as robust representation for time series forecasting. arXiv preprint arXiv:2402.05370.
> > > > > ##### [3] Lipton, Z. C., Kale, D. C., Elkan, C., & Wetzel, R. C. (2016, January). Learning to Diagnose with LSTM Recurrent Neural Networks. In ICLR (Poster).
> > > > > ##### [4] Harutyunyan, H., Khachatrian, H., Kale, D. C., Ver Steeg, G., & Galstyan, A. (2019). Multitask learning and benchmarking with clinical time series data. Scientific data, 6(1), 96.
> > > > > ##### [5] Preston, D., Protopapas, P., & Brodley, C. (2009, April). Event discovery in time series. In Proceedings of the 2009 SIAM International Conference on Data Mining (pp. 61-72). Society for Industrial and Applied Mathematics.
> > > > > ##### [6] Shchur, O., Türkmen, A. C., Januschowski, T., & Günnemann, S. (2021). Neural temporal point processes: A review. arXiv preprint arXiv:2104.03528.
> > > > > ##### [7] Ben-David, S., Blitzer, J., Crammer, K., Kulesza, A., Pereira, F., & Vaughan, J. W. (2010). A theory of learning from different domains. Machine learning, 79(1), 151-175.
> > > > > ##### [8] Koh, P. W., Sagawa, S., Marklund, H., Xie, S. M., Zhang, M., Balsubramani, A., ... & Liang, P. (2021, July). Wilds: A benchmark of in-the-wild distribution shifts. In International conference on machine learning (pp. 5637-5664). PMLR.

---

> > > > > > ### Comment · Reviewer_8fHo · 2025-08-07
> > > > > >
> > > > > > > Although reference points are globally shared, alignment is performed per sequence via timestamp-aware attention-based interpolation.
> > > > > >
> > > > > > According to Eq. 3, the quality of such interpolation is still bounded by the choice of global reference points and their learned embeddings. The time-series alignment issue, e.g., mismatch in the starting time of observations, could cause distribution shifts and invalidate the learned reference points and embeddings $\phi_h$. Thereby, the time-series alignment problem is highly relevant to the problem setup of ATENet and should be properly addressed for practical utility.

---

> > > > > > > ### Author Response · Authors · 2025-08-07
> > > > > > >
> > > > > > > Thank you for your follow-up. We would like to clarify our perspective on time-series misalignment and its relation to distribution shift in the context of our approach.
> > > > > > >
> > > > > > > First, start-time mismatch (or misalignment) is not inherently problematic in modern sequence modeling approaches such as Transformers or RNNs [1-6]. These models do not rely on absolute timestamps but instead learn representations based on the relative positions or temporal dynamics within each sequence. Therefore, the fact that sequences start at different times does not invalidate the model’s ability to learn meaningful patterns, nor does it by itself constitute a distribution shift.
> > > > > > >
> > > > > > > Second, in many real-world time-series domains, start times are often not entirely random. Instead, they tend to follow domain-specific routines or event triggers. For example, patient monitoring typically begins at symptom onset or ICU admission in clinical data, and measurements in industrial settings may align with batch starts or fault occurrences. Thus, even without explicit synchronization, there is often implicit regularity that the model can exploit. ATENet is designed to capture such latent patterns by learning reference points and generating representations aligned to them. Moreover, even in scenarios where start times are highly variable or nearly random, ATENet is still likely to work well by learning uniformly distributed reference points that provide broad temporal coverage without depending on strict alignment.
> > > > > > >
> > > > > > > We will revise the manuscript to clarify these points and prevent potential confusion regarding the role of alignment in our problem setting, while also highlighting the practical utility and robustness of ATENet in real-world deployments.
> > > > > > >
> > > > > > > ##### [1] Niu, P., Zhou, T., Wang, X., Sun, L., & Jin, R. (2024). Attention as robust representation for time series forecasting. arXiv preprint arXiv:2402.05370.
> > > > > > > ##### [2] Shaw, P., Uszkoreit, J., & Vaswani, A. (2018, June). Self-Attention with Relative Position Representations. In Proceedings of the 2018 Conference of the North American Chapter of the Association for Computational Linguistics: Human Language Technologies, Volume 2 (Short Papers) (pp. 464-468).
> > > > > > > ##### [3] Lohit, S., Wang, Q., & Turaga, P. (2019). Temporal transformer networks: Joint learning of invariant and discriminative time warping. In Proceedings of the IEEE/CVF Conference on Computer Vision and Pattern Recognition (pp. 12426-12435).
> > > > > > > ##### [4] Lipton, Z. C., Kale, D. C., Elkan, C., & Wetzel, R. C. (2016, January). Learning to Diagnose with LSTM Recurrent Neural Networks. In ICLR (Poster).
> > > > > > > ##### [5] Tallec, C., & Ollivier, Y. (2018, April). Can recurrent neural networks warp time?. In International Conference on Learning Representation 2018.
> > > > > > > ##### [6] Che, Z., Purushotham, S., Cho, K., Sontag, D., & Liu, Y. (2018). Recurrent neural networks for multivariate time series with missing values. Scientific reports, 8(1), 6085.

---

### Note · Authors · 2025-08-12

We sincerely thank the reviewers for their thoughtful and constructive feedback. We have carefully addressed all comments during the rebuttal phase and engaged in in-depth and productive discussions to clarify our contributions and methodology.

Regarding Reviewer 8fHo’s concern about start-time mismatch, we respectfully note that modern sequence models (e.g., Transformers and RNNs) primarily learn relative timing and dynamics rather than relying on a common absolute start time. As a result, moderate variation in sequence onset generally does not pose a problem. In many real-world settings, start times also follow implicit regularities that the model can learn. ATENet is designed with this in mind and remains robust when start times vary or are randomly offset. Accordingly, we believe that the start-time mismatch is a minor issue rather than a substantive limitation of our approach. In addition, regarding Reviewer x3TD’s question on hyperparameters, we reiterate that none were tuned on the test set and all were selected based on validation performance.

The reviewers’ suggestions have been invaluable, and we will incorporate these improvements and clarifications into the revised version. We appreciate the opportunity to engage in this dialogue and believe that the discussions have helped strengthen the quality and clarity of our work.

---

### Decision · Program_Chairs · 2025-09-17

**Decision:**

Accept (poster)

**Comment:**

This paper introduces a novel framework for irregular multivariate time-series classification. The method learns adaptive reference time points and incorporates two consistency regularizations to handle missingness and irregular sampling. Experiments show consistent improvements and better efficiency compared to a range of baselines.

Overall, this paper proposes a technically sound and well-presented framework for irregular multivariate time-series classification. The key contributions are the adaptive encoding of reference time points and the use of temporal/intervariable consistency regularizations, which systematically address missingness patterns and cross-variable dependencies. While the conceptual novelty and broader impact may not reach the highest bar for NeurIPS, the paper is technically solid, demonstrates consistent empirical improvements, and clearly articulates its methodological design. On balance, I recommend acceptance at the poster level.

The reviewers recognized the novelty of framing interpolation-based encoding for irregular time series, though some felt that the approach combines existing components in a way that may be perceived as incremental. One reviewer maintained concerns regarding generalization under distributional shifts, which the authors argued was outside the intended scope. Other reviewers were satisfied with the authors’ clarifications on hyperparameter tuning, baseline selection, and interpolation robustness.